# Eukaryotic translation initiation factor 3 plays distinct roles at the mRNA entry and exit channels of the ribosomal preinitiation complex

Colin Echeverría Aitken[1], Petra Beznosková[2], Vladislava Vlčková[2], Wen-Ling Chiu[3†], Fujun Zhou[1], Leoš Shivaya Valášek[2*], Alan G Hinnebusch[3*], Jon R Lorsch[1*]

[1]Laboratory on the Mechanism and Regulation of Protein Synthesis, Eunice Kennedy Shriver National Institute of Child Health and Human Development, National Institutes of Health, Bethesda, United States; [2]Laboratory of Regulation of Gene Expression, Institute of Microbiology ASCR, Prague, Czech Republic; [3]Laboratory of Gene Regulation and Development, Eunice Kennedy Shriver National Institute of Child Health and Human Development, National Institutes of Health, Bethesda, United States

*For correspondence: valasekl@
biomed.cas.cz (LSV);
ahinnebusch@nih.gov (AGH); jon.
lorsch@nih.gov (JRL)

Present address: †PharmaEssentia Corp., Taipei, Taiwan

**Abstract** Eukaryotic translation initiation factor 3 (eIF3) is a central player in recruitment of the pre-initiation complex (PIC) to mRNA. We probed the effects on mRNA recruitment of a library of *S. cerevisiae* eIF3 functional variants spanning its 5 essential subunits using an in vitro-reconstituted system. Mutations throughout eIF3 disrupt its interaction with the PIC and diminish its ability to accelerate recruitment to a native yeast mRNA. Alterations to the eIF3a CTD and eIF3b/i/g significantly slow mRNA recruitment, and mutations within eIF3b/i/g destabilize eIF2•GTP•Met-tRNA$_i$ binding to the PIC. Using model mRNAs lacking contacts with the 40S entry or exit channels, we uncovered a critical role for eIF3 requiring the eIF3a NTD, in stabilizing mRNA interactions at the exit channel, and an ancillary role at the entry channel requiring residues of the eIF3a CTD. These functions are redundant: defects at each channel can be rescued by filling the other channel with mRNA.

## Introduction

The goal of translation initiation is to assemble the ribosome at the start (AUG) codon of an mRNA in a state ready to begin reading the message and synthesizing the corresponding protein. This represents one of the most important readings of the genetic code because initiation at the wrong site results in extended, truncated or out of frame translation products. To ensure faithful identification of the start codon, translation initiation in eukaryotes follows a complex, multi-step pathway which requires the participation of 12 initiation factors (eIFs) (reviewed in: (*Aitken and Lorsch, 2012*; *Hinnebusch, 2014*; *Jackson et al., 2010*; *Sonenberg and Hinnebusch, 2009*; *Valásek, 2012*). The process begins with the formation of a 43S pre-initiation complex (PIC), in which the initiator methionyl tRNA (Met-tRNA$_i$) is initially positioned as a tRNA$_i$•eIF2•GTP ternary complex (TC) on the small (40S) ribosomal subunit. The 43S PIC further contains eIF1 and eIF1A bound, respectively, near the 40S P and A sites, as well as eIF5 and eIF3. GTP hydrolysis by eIF2 can proceed within the PIC, but the release of inorganic phosphate (P$_i$) is blocked by the presence of eIF1. With the collaboration of eIF4B and the eIF4F complex composed of eIF4A, eIF4G, and eIF4E, the PIC is then loaded onto the

**eLife digest** Cells use the genetic information stored within genes to build proteins, which are largely responsible for performing the molecular tasks essential for life. The ribosome is the molecular machine that translates the information within genes to assemble proteins in all cells, from bacteria to humans. To make a protein, the corresponding gene is first copied to make molecules of messenger ribonucleic acid (or mRNA for short). Then the ribosome binds to the mRNA in a process called translation initiation.

Cells tightly regulate translation initiation so that they can decide which proteins to make, according to their needs and in response to changes in the environment. In fact, regulation of translation initiation is often disrupted during viral infections, cancer and other human diseases. A set of proteins called translation initiation factors drive translation initiation; the largest and least understood of these is called eIF3.

Cells are unable to load the mRNA onto the ribosome without eIF3, which has two "arms" that sit near where the mRNA enters and exits the ribosome. Aitken et al. used mutant forms of eIF3 from genetically modified yeast to investigate how the arms of the protein work, and if they help the ribosome hold onto the mRNA.

These experiments show that the two arms of eIF3 have unique roles. One arm sits near where mRNA exits the ribosome and is important for holding onto the mRNA. The other arm – which is near where mRNA enters the ribosome – helps hold the ribosome and other components of the translation machinery together. This arm may also help to open and close the channel through which messenger RNA enters the ribosome. The next challenges are to find out the precise role this arm plays in translation – in particular, how it helps to open and close the channel in the ribosome, and whether this helps the ribosome load the messenger RNA or even move along it.

5' end of the mRNA, from which it scans in the 3' direction to locate the AUG codon, thus generating the 48S PIC. Both loading onto the mRNA and subsequent scanning are facilitated by an open conformation of the PIC enforced by the binding of eIF1. Recognition of the AUG codon triggers eviction of eIF1, gating $P_i$ release by eIF2 and provoking a transition from the open PIC conformation to a closed conformation that arrests scanning. Joining of a 60S ribosomal subunit completes assembly of the 80S initiation complex, which can then proceed to the elongation cycle.

The largest of the initiation factors is eIF3, a multisubunit complex that binds the 40S subunit and has been implicated in events throughout the initiation pathway (*Hinnebusch, 2006*, *2011*; *Valásek, 2012*). In humans, eIF3 is composed of 13 individual subunits. In the yeast *S. cerevisiae*, eIF3 comprises 5 essential subunits thought to represent a core complex capable of performing the essential tasks of eIF3: eIF3a, eIF3b, eIF3c, eIF3i, and eIF3g. A sixth nonessential and nonstoichiometric subunit, eIF3j, associates loosely with the complex (*Elantak et al., 2010*; *Fraser et al., 2004*; *Nielsen et al., 2006*; *Valasek et al., 2001a*), though its role during initiation remains unclear (*Beznosková et al., 2013*; *Fraser et al., 2007*; *Mitchell et al., 2010*). Genetic and biochemical evidence indicate that eIF3 stabilizes both the 43S (*Asano et al., 2000*; *Kolupaeva et al., 2005*; *Maag et al., 2005*; *Valášek et al., 2003*) and 48S PIC (*Chiu et al., 2010*; *Khoshnevis et al., 2014*; *Phan et al., 2001*) and interacts with TC (*Valášek et al., 2002*), eIF1 (*Fletcher et al., 1999*; *Valasek et al., 2004*), eIF1A (*Olsen et al., 2003*), and eIF5 (*Asano et al., 2001*; *Phan et al., 1998*), as well as with the 40S subunit near both the mRNA entry and exit channels (*Kouba et al., 2012a*, *2012b*; *Pisarev et al., 2008*; *Valášek et al., 2003*). eIF3 also plays roles in loading the mRNA onto the PIC (*Jivotovskaya et al., 2006*; *Mitchell et al., 2010*; *Pestova and Kolupaeva, 2002*) and in scanning of the mRNA to locate the start codon (*Chiu et al., 2010*; *Cuchalova et al., 2010*; *Karásková et al., 2012*; *Nielsen et al., 2006*; *Valasek et al., 2004*). How eIF3 contributes to these diverse events is not yet clear.

Recent results have highlighted the role of eIF3 in mRNA recruitment to the PIC. The depletion of eIF3 from yeast cells sharply decreases the amount of mRNA associated with PICs in vivo (*Jivotovskaya et al., 2006*). In vitro, yeast eIF3 is required for detectable recruitment of native mRNAs in the yeast reconstituted translation initiation system (*Mitchell et al., 2010*). Mutations in

the C-terminal domain (CTD) of eIF3a, which interacts with the 40S subunit near the mRNA entry channel, impair mRNA recruitment in yeast cells without affecting the integrity of the PIC (*Chiu et al., 2010*). The eIF3a N-terminal domain (NTD) interacts functionally with mRNA near the exit channel, enhancing reinitiation upon translation of the upstream open reading frame 1 (uORF1) and uORF2 of *GCN4* mRNA in a manner dependent on the sequence upstream of these two uORFs (*Gunišová et al., 2016*; *Gunišová and Valášek, 2014*; *Munzarová et al., 2011*; *Szamecz et al., 2008*). Consistent with this, mammalian eIF3 can be cross-linked to mRNA at positions 8–17 nucleotides upstream of the AUG codon (*Pisarev et al., 2008*), near the exit channel, and a mutation within the eIF3a NTD, which has been located near the exit channel in structures of the PIC, interferes with mRNA recruitment in yeast cells (*Khoshnevis et al., 2014*).

How eIF3 might intervene at both the mRNA entry and exit channels of the PIC was illuminated recently by a series of structural studies. Initial structural information — based on separate lower resolution Cryo-EM reconstructions — placed mammalian eIF3 on the solvent-exposed face of the 40S subunit, near the platform, but did not reveal interactions in either the entry or exit channel (*Siridechadilok et al., 2005*). Subsequent studies in the presence of the helicase DHX29 refined this placement, identifying two main points of contact near the 40S platform, as well as potential peripheral domains of eIF3 near the 40S head above the platform and below DHX29 nearer the mRNA entry channel (*Hashem et al., 2013*).

More recently, high-resolution Cryo-EM reconstructions of mammalian eIF3 bound to a PIC containing DHX29 (*Georges, des et al., 2015*) and of yeast eIF3 bound either to a 40S•eIF1•eIF1A complex (*Aylett et al., 2015*) or to a partial 48S PIC containing mRNA (*Llácer et al., 2015*) (*Figure 1A–C*, py48S-closed structure) have revealed this interaction in molecular detail. In these structures, eIF3 (shown in blues and reds from py48S-closed structure) is found bound primarily to the 40S (light grey) solvent face, via two main points of contact. First, the eIF3a (dark blue) NTD and eIF3c (light blue) CTD PCI (**P**roteasome/**C**op9/e**I**F3) domains form a heterodimer that binds below the platform, near the mRNA exit channel pore (*Figure 1B*). A second contact is formed closer to the mRNA entry-channel side of the PIC (*Figure 1A*), in the vicinity of helix 16 (h16) and helix 18 (h18) of the 40S rRNA. This contact is made by the eIF3b (light blue) β-propeller domain, consistent with its described interaction with uS4 (*Liu et al., 2014*), and is likely connected to the first contact via a previously identified interaction between the eIF3b RRM and the eIF3a CTD (*Valasek et al., 2001b*; *Valášek et al., 2002*), which likely spans the solvent face (*Figure 1C*). In both the mammalian structure and the yeast eIF3•40S•eIF1•eIF1A structure, the eIF3i and eIF3g subunits are found above the eIF3b subunit, near the solvent-face opening of the mRNA entry channel; with the exception of the non-essential eIF3j subunit, no density for eIF3 was observed on the intersubunit face of the 40S in these structures. In the partial 48S structure (*Figure 1B*), the eIF3c (light blue) NTD extends from the eIF3a/eIF3c PCI heterodimer near the exit channel towards the intersubunit face, projecting near the P-site, where it appears to interact with eIF1 (pink). Near the entry channel, the eIF3a CTD extends away from eIF3b (*Figure 1A*), wrapping around the 40S, spanning the intersubunit face, and nearly meeting the eIF3c NTD near the P-site, effectively encircling the 43S PIC (*Figure 1C*). In this structure, the eIF3i (maroon) and eIF3g (red) subunits are located together with the eIF3b C-terminal region on the intersubunit face near the A site and below the intersubunit-face opening of the mRNA entry channel, where eIF3i interacts with the eIF2γ portion of the TC (green and charcoal). In a recent structure of mammalian eIF3 bound to an apparently late-stage initiation complex, eIF3i was found at a distinct location on the intersubunit face of the PIC, leading to the suggestion that the density observed near TC in the yeast py48S-closed structure is in fact the β-propeller of eIF3b and not eIF3i (*Simonetti et al., 2016*).

While these recent structures illuminate several important contacts between eIF3 and the PIC, the identification of less well-resolved domains of eIF3, as well as the proposed placement of unresolved regions of the complex rest heavily on a host of interactions previously identified in genetic and biochemical studies. These studies established interactions between the eIF3a CTD and the eIF3b RRM *Valasek et al., 2001b*), the 40S latch components uS3/uS5 and h16/18 (*Chiu et al., 2010*; *Valášek et al., 2003*), and with TC at its extreme C-terminus (*Valášek et al., 2002*). The C-terminal region of eIF3b was previously shown to be required for binding of eIF3i and eIF3g (*Asano et al., 1998*; *Herrmannová et al., 2012*), the latter of which was observed to interact with uS3 and uS10 (*Cuchalova et al., 2010*). These observations delineate a network of interactions between the eIF3a CTD, eIF3b, eIF3i, and eIF3g, and place them either in the vicinity of the entry channel or projecting

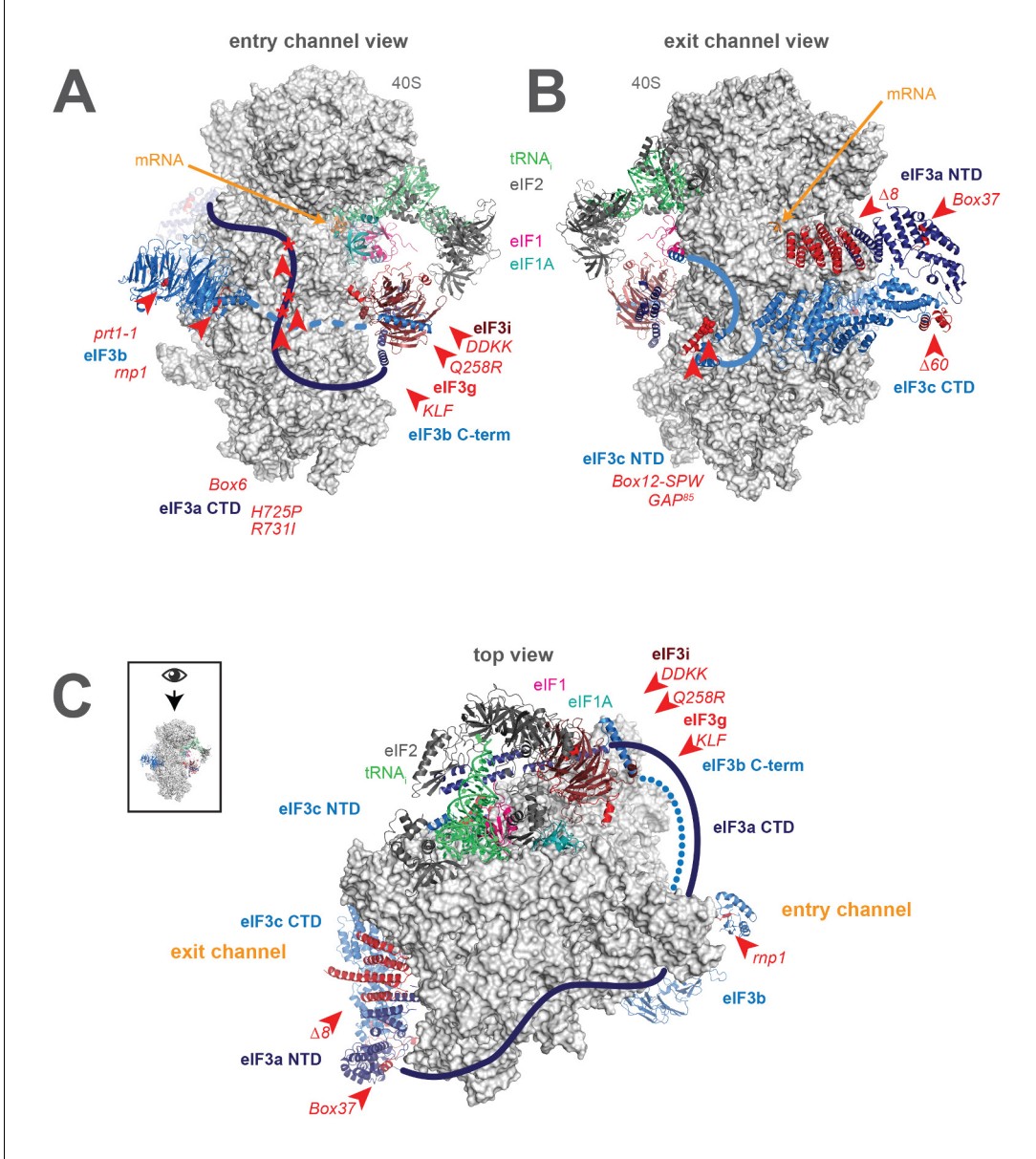

**Figure 1.** A library of *S. cerevisiae* eIF3 functional variants affected by mutations spanning all five essential subunits. Three views of the cryo-EM structure of eIF3 bound to the partial yeast 48S PIC in the closed conformation (py48S-closed), as reported by Llacer, *et al*. The 40S subunit is shown in light grey, with resolved domains of eIF3 shown as ribbons and unresolved regions cartooned as solid or dotted lines: eIF3a is shown in dark blue, eIF3c and eIF3b in light blue, eIF3i in maroon, and eIF3g in red. The location of mutations throughout eIF3 are shown colored in red and indicated by red arrowheads; the eIF3i and eIF3g subunits are also depicted in red hues owing to their absence in the a/b/c subcomplex resulting from the *DDKK* mutation in eIF3i. The mRNA (orange), Met-tRNA<sub>i</sub> (green), eIF2 (charcoal), eIF1 (pink), and eIF1A (teal) are also visible in this structure. (**A**) View looking towards the mRNA entry channel, with the 40S intersubunit face on the right, and the solvent face on the left. (**B**) Opposite view looking towards the mRNA exit channel, with the 40S intersubunit face on the left, and the solvent face on the right. (**C**) View of the PIC looking down at the 40S head, showing the relative orientation of the entry- and exit-channel arms of eIF3 and its contacts on the 40S solvent face. The PIC is oriented such that the intersubunit face appears at the top, while the solvent face appears at the bottom.

towards the 40S intersubunit face. Other studies identified interactions between the eIF3a NTD and uS2 (*Valášek et al., 2003*; *Kouba et al., 2012b*), the eIF3c CTD and RACK1 (*Kouba et al., 2012a*), and between the eIF3c NTD and both eIF5 and eIF1 (*Asano et al., 2000*; *Valasek et al., 2004*). This final observation, together with the discovery of mutations in the eIF3c NTD that affect the fidelity

of start-codon recognition (*Valasek et al., 2004*; *Karásková et al., 2012*), supports the putative identification of regions of density near the P site in the py48S-closed structure (*Llácer et al., 2015*).

Together with this previous work, structural studies reveal the presence of eIF3 not just on the 40S solvent face, but near the functional centers of the PIC. In particular, eIF3 appears to project arms into or near both the mRNA entry and exit channels (*Figure 1C*). Based on their apparent locations near either the solvent-face or intersubunit-face openings of the mRNA entry channel, eIF3i, eIF3g, the C-terminal region of eIF3b, and the eIF3a CTD together compose the entry-channel arm. The exit-channel arm comprises the eIF3a NTD and eIF3c CTD PCI heterodimer located below the platform at the solvent-face opening of the mRNA exit channel. The presence of eIF3 subunits at both mRNA channels, near the P site, and encircling the PIC does not, however, clarify the mechanistic nature of their involvement during steps of the initiation pathway. These regions of eIF3 might participate actively in these events, interacting and collaborating with the 40S, mRNA, and initiation factors to stimulate mRNA recruitment, scanning, and start-codon recognition. eIF3 might also alter the conformation of the PIC to indirectly enhance the efficiency of these steps or serve as a scaffold upon which other factors can bind.

To interrogate the mechanistic roles of the domains of eIF3, we purified a library of *S. cerevisiae* eIF3 variants – all of which were previously characterized in vivo (*Chiu et al., 2010*; *Cuchalova et al., 2010*; *Herrmannová et al., 2012*; *Karásková et al., 2012*; *Khoshnevis et al., 2014*; *Kouba et al., 2012a*; *Nielsen et al., 2004*, *2006*; *Szamecz et al., 2008*) – to probe their effects on steps and interactions in the translation initiation pathway in an in vitro-reconstituted yeast translation system (*Acker et al., 2007*; *Algire et al., 2002*; *Mitchell et al., 2010*). This library contains variants with mutations spanning the five essential subunits of eIF3 (*Table 1*). The phenotypes associated with these mutants suggest defects in 40S binding, 43S PIC formation, mRNA recruitment, scanning, and start-codon recognition. Our results demonstrate that regions throughout the eIF3 complex contribute to its interaction with the PIC and its role in mRNA recruitment. In particular, the mRNA entry-channel arm of eIF3 and eIF3b appear to play a role in stabilizing TC binding to the PIC and accelerating mRNA recruitment. In contrast, the mRNA exit channel arm of eIF3 is required to stabilize the binding of mRNA within the PIC, and this function can be attributed to the extreme N-terminal region of the eIF3a subunit.

## Results

The eIF3 variants in this study can be divided into four categories (*Table 1*). The first contains mutations in the components of the eIF3 entry channel arm (*Figure 1A*). This includes mutations to the eIF3i (*Q258R, DDKK*) and eIF3g (*KLF*) subunits (*Asano et al., 1998*; *Cuchalova et al., 2010*; *Herrmannová et al., 2012*). The eIF3i *DDKK* mutation disrupts the interaction between the eIF3i/g module and eIF3b, resulting in the dissociation of eIF3i/g from the sub-complex containing the a, b, and c subunits (*Herrmannová et al., 2012*); eIF3 purified from *DDKK* cells yields the WT eIF3a/b/c sub-complex. These mutations produce phenotypes consistent with scanning defects, and in the case of the *DDKK* mutation also reduce binding of eIF3 to the 43S PIC in vivo. Also included in this group are mutations to the eIF3a CTD (*H725P, R731I, Box6*) (*Valasek et al., 1998*; *Chiu et al., 2010*), which has not been resolved in recent structural models but was shown to interact with the RRM of eIF3b (*Valasek et al., 2001b*), as well as with 40S components near the mRNA entry channel: uS5, uS3, h16, and h18 (*Chiu et al., 2010*; *Valášek et al., 2003*). These mutations affect mRNA recruitment, scanning, and codon recognition in vivo, perhaps by affecting the equilibrium between the open and closed states of the 43S PIC.

The second group of mutations affects residues of eIF3b located at the solvent face of the 40S subunit (*Figure 1A*). The *prt1-1* mutation was the first functional variant of eIF3 to be identified (*Evans et al., 1995*; *Hartwell and McLaughlin, 1969*) and is a point-substitution (S555F) within the eIF3b β-propeller domain that contacts the 40S subunit and is connected to the eIF3i/g module of the entry-channel arm via the C-terminal segment of eIF3b (*Herrmannová et al., 2012*). The *rnp1* mutation is within the eIF3b N-terminal RRM domain (*Nielsen et al., 2006*), which interacts with the eIF3a CTD portion of the entry-channel arm (*Valasek et al., 2001b*). Both eIF3b mutations give rise to a complex portfolio of phenotypes consistent with defects in 40S binding, 43S PIC formation, scanning, and codon recognition (*Nielsen et al., 2004*, *2006*).

**Table 1.** Putative mechanistic defects attributed to eIF3 variants. Table describing the putative mechanistic defects attributed to each eIF3 variant within our library. Variants are organized according to their general location within the PIC (left column) and by the subunit or subunit domain (second column from left) in which the associated mutation occurs (allelic designation in third column from left). The specific identity of the mutation affecting each variant (second column from right) and the putative mechanistic defects previously attributed to each variant (final column, see text for references) are described.

| Location | Subunit | Variant | Mutation | Putative defects |
|---|---|---|---|---|
| Entry channel | eIF3a/Tif32 CTD | H725P | H725P | mRNA recruitment scanning codon recognition |
| | | R731I | R731I | |
| | | Box6 | Alanine substitution of $^{692}$LDLDTIKQVI$^{701}$ | |
| | eIF3i/Tif34 | Q258R | Q258R | scanning |
| | | DDKK | D207K/D224K | 40S binding 43S PIC formation scanning start codon recognition |
| | eIF3g/Tif35 | KLF | K194A/L235A/F237A | scanning |
| Solvent face | eIF3b/Prt1 | prt1-1 | S555F | 40S binding 43S PIC formation scanning start codon recognition |
| | | rnp1 | Alanine substitution of $^{124}$KGFLVE$^{130}$ | |
| Exit channel | eIF3c/Nip1 CTD | Δ60 | C-terminal truncation of aa 753–813 | 40S binding 43S PIC formation |
| | eIF3a/Tif32 NTD | Δ8 | N-terminal truncation of aa 1–200 | 40S binding 43S PIC formation re-initiation |
| | | Box37 | Alanine substitution of $^{361}$PTRKEMLQSI$^{370}$ | mRNA recruitment |
| P site | eIF3c/Nip1 NTD | GAP$^{85}$ | Deletion of aa 60–144 | eIF1 binding start codon recognition |
| | | Box12 -SPW | K111S/K114P/K116W | |

The next group includes mutations to regions of eIF3 located near the mRNA exit channel of the 40S ribosomal subunit (*Figure 1B*). Among these are a 200 amino acid N-terminal truncation of the eIF3a PCI domain (*Δ8*), as well as a multiple alanine substitution deeper within the eIF3a PCI domain (*Box37*). The *Δ8* mutation disrupts a previously identified interaction between eIF3a and uS2 (*Valášek et al., 2003*), apparently weakening the eIF3:40S interaction (*Szamecz et al., 2008*), whereas the *Box37* substitution predominantly reduces mRNA association with native PICs in vivo (*Khoshnevis et al., 2014*). The final mutant in this group is a 60 amino acid C-terminal truncation of the eIF3c PCI domain (*Δ60*) that reduces the amount of both eIF3 and eIF5 in 43S PICS, apparently by disrupting interactions with the ribosome-associated protein RACK1 and the 18S rRNA (*Kouba et al., 2012a*).

Lastly, we investigated mutations within the N-terminal region of eIF3c (*GAP$^{85}$, Box12-SPW*) that is thought to project near the P site and mediate interactions with eIF1 near the decoding center of the 40S subunit (*Figure 1B*). These mutations affect the amount of eIF1 bound to 43S PICs and appear to interfere with stringent selection of the AUG start-codon (*Karásková et al., 2012*).

## Both the mRNA entry and exit channel arms of eIF3 and the eIF3b subunit contribute to stabilizing eIF3 interaction with the 43S complex

Genetic approaches have identified mutations in eIF3 that reduce its abundance in native 43S or 48S PICs in vivo, suggesting the altered residues are involved in interactions with the 40S ribosomal

subunit or other components of the PIC. This interpretation is corroborated by structural studies that identify points of contact between the 43S PIC and eIF3, as described above. To directly interrogate the contributions of the eIF3 subunits to the 43S:eIF3 interaction, we first examined the ability of each eIF3 variant to bind 43S complexes in vitro. To this end, we assembled 43S complexes using purified 40S subunits, saturating levels of eIF1 and eIF1A, and a limiting concentration of TC containing initiator tRNA$_i$$^{Met}$ charged with [$^{35}$S]-methionine, and incubated them with increasing concentrations of each variant (*Figure 2A,B*). Native gel electrophoresis of the reactions enables separation of free tRNA$_i$ from TC bound to both 43S and 43S•eIF3 complexes. In the absence of AUG-containing mRNA, and at the low concentrations of 40S subunits required to measure the affinity of eIF3 for the 43S complex, there is a significant population of free 40S subunits that can compete for eIF3 and complicate the measurement of 43S•eIF3 complex formation. However, because all TC-containing species are resolved by this method, the amount of free eIF3 (as opposed to total eIF3) can be calculated from measured dissociation constants, enabling us to determine the affinity of eIF3 for the 43S complex under these conditions (*Figure 2—figure supplement 1*).

Both wild-type (WT) and variant forms of eIF3 exhibit apparently cooperative binding to the 43S PIC, as evidenced by sigmoidal binding isotherms (*Figure 2—figure supplement 2A*). This behavior may reflect the requirement for stable interaction between the five subunits of the complex prior to binding the PIC. We modeled binding data using both a standard Langmuir binding isotherm and the cooperative Hill curve, and obtained similar results with both methods (*Figure 2—figure supplement 2B*). Because fitting with the Hill equation more accurately models the data (*Figure 2—figure supplement 2A*), those results are discussed below. The Hill constants resulting from the fits are measures of apparent cooperativity, but we cannot yet ascribe a molecular interpretation to them.

WT eIF3 binds tightly to the 43S complex, with a dissociation constant ($K_D$) near or below ($\leq$38 nM; *Figure 2C*, left panel) the concentration of 40S subunits used in the experiment (30 nM). Mutations in the entry channel arm of eIF3 appear to weaken its binding by up to ~3 fold (*Figure 2C*, left panel). Two of the eIF3a CTD mutations in this region, *H725P* and *R731I*, which alter the KERR motif of the eIF3a CTD Hcr1-like Domain (HLD), decrease the apparent affinity of eIF3 for the 43S PIC by ~2 and ~3 fold, respectively. The *Box6* mutant, which is also located within the eIF3a HLD but is outside the KERR motif, does not affect the apparent affinity of eIF3 for the 43S complex. The a/b/c sub-complex lacking the i and g subunits exhibits ~2 fold weaker apparent affinity for the 43S PIC than the WT factor, whereas the eIF3i *Q258R* and the eIF3g *KLF* mutations do not manifest any effect on the 43S:eIF3 interaction. In contrast, the *prt1-1* and *rnp1* mutations in eIF3b, which connects the entry channel arm to the 40S solvent face, also weaken the 43S:eIF3 affinity ~2 and ~3 fold, respectively.

We observe similarly modest effects in the case of the exit-channel mutants of the eIF3a NTD, the *Δ8* truncation and *Box37* substitution. Both weaken the apparent 43S:eIF3 affinity by ~2 fold or less. The *Δ60* mutation within the eIF3c PCI domain, which is also located near the exit channel, does not affect the 43S:eIF3 affinity. Nearer to the P site, the *Box12-SPW* mutation in the N-terminal region of eIF3c weakens the 43S:eIF3 interaction ~2 fold, whereas the *GAP$^{85}$* mutation also located within the eIF3c NTD does not appear to affect the binding of eIF3 to 43S PICs.

To examine the effects of these mutations on the affinity of eIF3 for 48S PICs, we repeated these experiments in the presence of saturating amounts of an unstructured, AUG-containing model mRNA (*Figure 2C*, right panel). As with our 43S PIC binding experiments, these experiments produced apparently cooperative binding behavior. WT eIF3 binds 48S PICs tightly ($K_D \leq 38$ nM), similar to what we observed for 43S PICs lacking mRNA. The absence of the i and g subunits in the a/b/c sub-complex weakens the apparent eIF3:48S interaction by nearly three-fold, as do the *prt1-1* and *rnp1* mutations in eIF3b, by nearly three-fold and more than four-fold, respectively. We also observe ~2 fold reductions in apparent eIF3 affinity for 48S complexes with mutations in the exit channel arm of eIF3 (*Δ8* and *Box37*) and the *Box12-SPW* mutation in the N-terminal region of eIF3c. In fact, the apparent affinity of most of the eIF3 variants for 48S PICs is similar to that observed for 43S PICs lacking mRNA, perhaps because their apparent 43S PIC affinity, even in the absence of mRNA, is tighter than the resolution of our assay. An alternative explanation is that stabilizing contacts with the mRNA are offset by changes in the complex that weaken other interactions with eIF3. In contrast, both the *H725P* and *R731I* substitutions within the eIF3a CTD display higher apparent affinities in the presence of mRNA than in its absence, perhaps because mRNA contributes interactions with eIF3 that can supplement those disrupted by these mutations. It is also possible that both

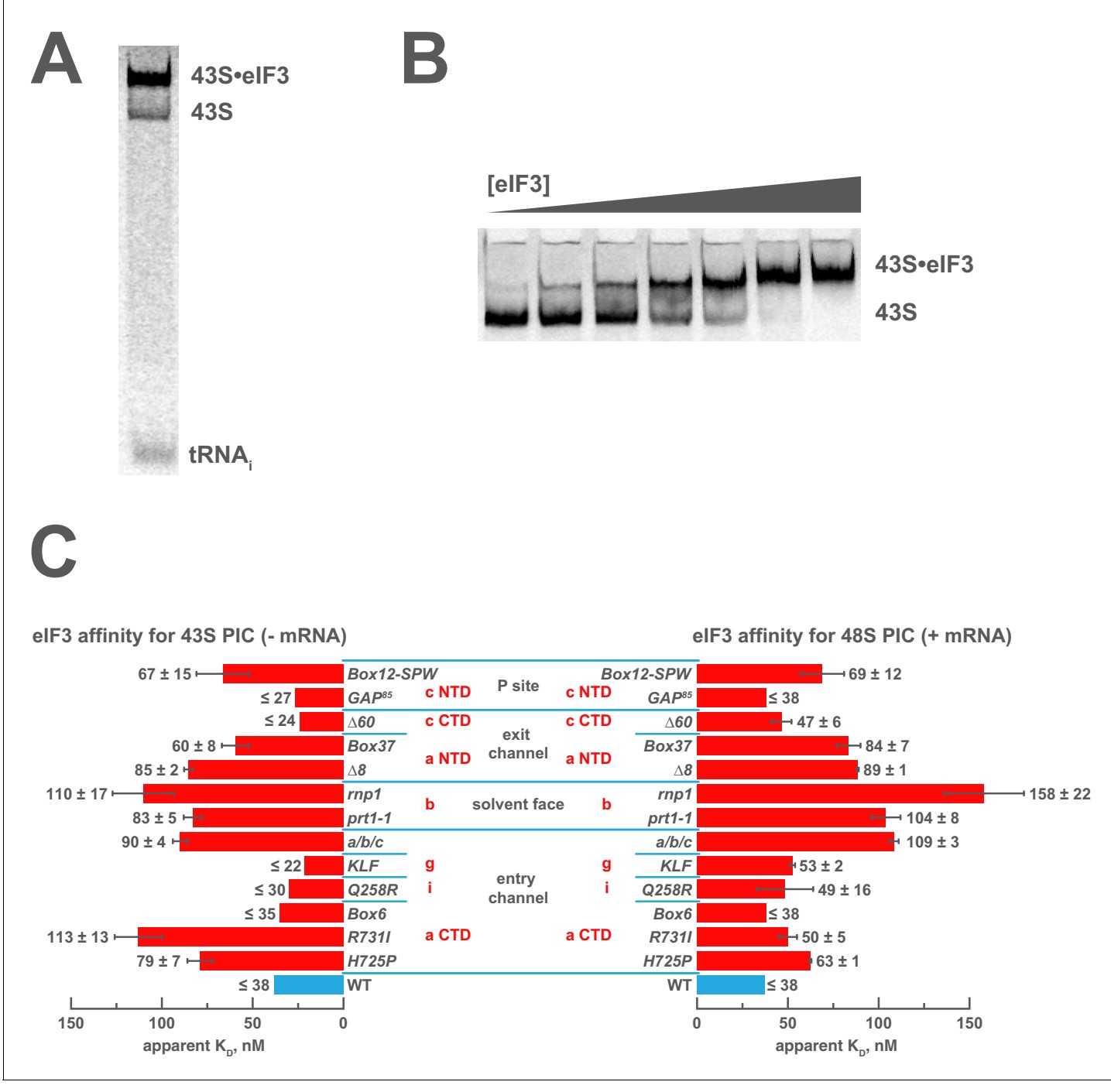

**Figure 2.** Mutations in the mRNA entry- or exit-channel arms of eIF3 or in eIF3b destabilize eIF3 binding to the PIC. (**A**) Binding of ternary complex containing [$^{35}$S]-Met-tRNA$_i^{Met}$ to 40S ribosomal subunits in the presence of eIF1, eIF1A, and WT or mutant eIF3 was measured using a native gel-shift assay, which separates free [$^{35}$S]-Met-tRNA$_i^{Met}$ from that bound to 43S PICs alone or 43S PICs containing eIF3 (43S·eIF3). (**B**) The titration of eIF3 into reactions containing 43S PICs produces a well-resolved gel-shift that monitors the binding of eIF3 to the PIC. The amounts of [$^{35}$S]-Met-tRNA$_i^{Met}$ free, bound to 43S PICs, and bound to 43S·eIF3 complexes were quantified and analyzed as described in *Figure 2—figure supplement 1*. (**C**) The apparent equilibrium dissociation constant ($K_D$) of WT (blue bars) and eIF3 variants (red bars) for the PIC, both in the absence (left) and presence (right) of mRNA, obtained by fitting the data with the Hill equation (see *Figure 2—figure supplement 2B*). Bars and errors represent mean and SEM, respectively (as determined by individual fitting of each experiment), of ≥2 experiments. Owing to the conditions of these assays, apparent affinities ≤ 30 nM likely represent upper limits.

The following source data and figure supplements are available for figure 2:

*Figure 2 continued on next page*

*Figure 2 continued*

**Source data 1.** Individual eIF3:43S and eIF3:48S dissociation constant measurements.
**Figure supplement 1.** Calculating the affinity of eIF3 for PICs under conditions where free 40S subunits compete for binding.
**Figure supplement 2.** Comparison of hyperbolic and Hill equations for modeling eIF3 binding to the PIC.

variants, which appear to affect the transition between open and closed conformations of the PIC in vivo (*Chiu et al., 2010*), bind more readily to the closed conformation of the PIC stabilized by AUG-containing mRNA. Presumably, the contribution of these effects to the affinity of WT eIF3 for 48S complexes is masked by its already tight interaction with the 43S PIC lacking mRNA.

Nonetheless, these results suggest that both the eIF3 entry- and exit-channel arms, together with eIF3b, contribute to stabilizing eIF3 binding to the PIC. This is consistent with structural and genetic evidence identifying multiple points of contact between eIF3 and the 40S subunit (*Aylett et al., 2015*; *Chiu et al., 2010*; *Llácer et al., 2015*; *Valášek et al., 2003*). Accordingly, impairing any single interaction with these mutations does not dramatically destabilize eIF3 binding to the 43S PIC.

## The eIF3 entry channel arm and eIF3b stabilize TC association with the 43S PIC

We next interrogated the contributions of the eIF3 subunits to stabilizing TC binding to the PIC by measuring the dissociation constant of TC in 43S complexes assembled with each eIF3 variant. These experiments employed the assay described above, only each eIF3 variant was included at saturating levels relative to its measured affinity for the 43S PIC and the concentration of 40S subunits was varied. We have previously reported the thermodynamic coupling between TC binding to the 43S complex and the presence of mRNA with an AUG codon in the 40S P site. In the presence of AUG-containing model mRNA, the affinity of TC for the 43S PIC is ~10 pM (*Kolitz et al., 2009*); in the absence of mRNA this affinity is reduced by 4 orders of magnitude to 104 ± 15 nM (*Figure 3*, left panel). WT eIF3 stabilizes TC binding to the 43S PIC by ~7 fold, resulting in an affinity of 15 ± 1 nM in the absence of mRNA (*Figure 3*, left panel). For most eIF3 variants, this affinity is not significantly altered. However, the a/b/c sub-complex lacking the i and g subunits weakens the 43S:TC interaction ~ 3 fold, as does the *prt1-1* mutation in the eIF3b β-propeller domain. The eIF3b RRM mutation *rnp1* has a slightly smaller effect on 43S:TC affinity, weakening it ~ 2 fold (*Figure 3*, left panel).

Intriguingly, three mutations – *R731I* and *Box6* in the eIF3a CTD and *GAP85* in the eIF3c NTD – appear to modestly increase the affinity of TC relative to the effect of WT eIF3. Several mutations also reduce the endpoints of TC recruitment observed at saturating concentrations of 40S subunits: *Box12-SPW*, *rnp1* and *R731I* (*Figure 3*, right panel). Previous work has suggested that such reductions may be due to a shift in the balance of complexes between two conformational states: one in which TC is stably bound and one in which it dissociates during gel electrophoresis (*Kapp et al., 2006*; *Kolitz et al., 2009*). It is possible that these states are related to the open and closed conformations of the PIC and that eIF3 modulates their relative stabilities.

Our results suggest that segments of eIF3b, the eIF3a CTD, and eIF3 i/g module – which constitute the eIF3 entry-channel arm and its attachment point on the 40S solvent face – contribute, together with the eIF3c NTD, to stabilizing TC binding to the 43S PIC, perhaps by modulating the PIC conformation or by interacting indirectly or directly with TC. The latter explanation would be consistent with the structure of the py48S-closed complex, in which the i and g subunits are found on the intersubunit face, resting against eIF2γ (*Llácer et al., 2015*) The effects of the eIF3c NTD mutants *GAP85* and *Box12-SPW* might occur as a result of indirect interactions with TC, consistent with their effects on eIF1 binding (*Karásková et al., 2012*).

## Regions throughout eIF3 collaborate to drive mRNA recruitment

To dissect the contributions of regions throughout eIF3 to mRNA recruitment, we followed the kinetics and extent of recruitment in the presence of each eIF3 variant in vitro. For those variants that

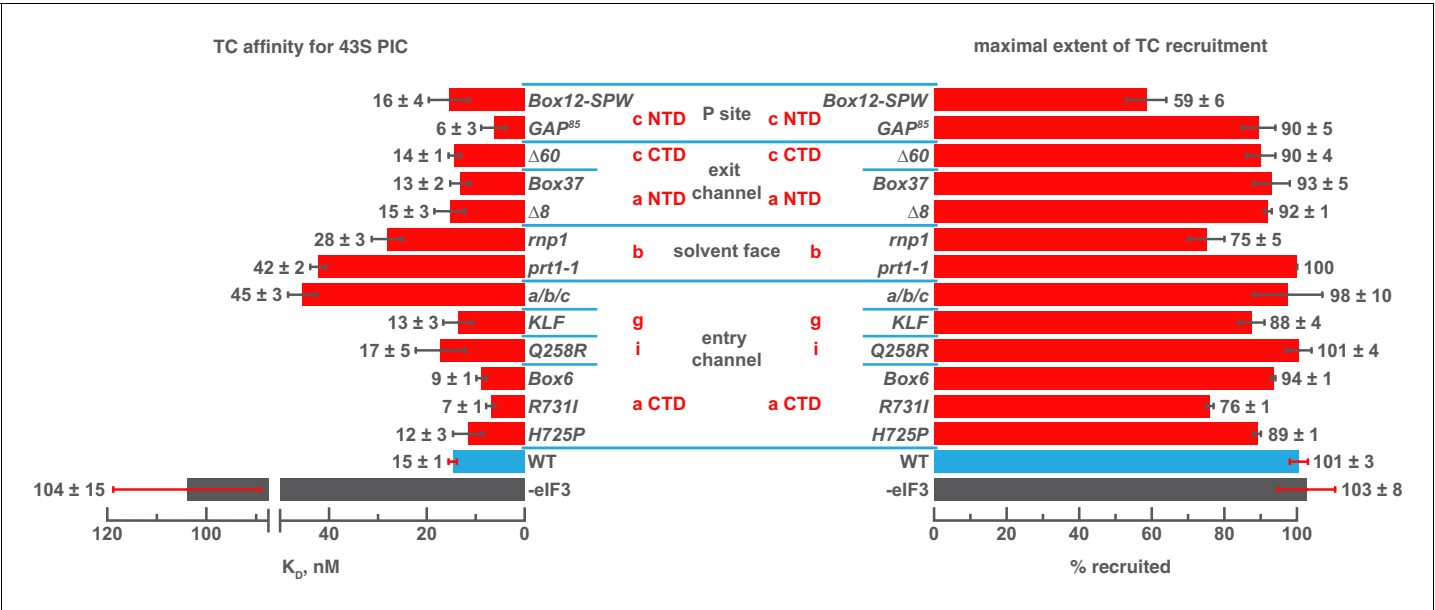

**Figure 3.** Mutations in the mRNA entry-channel arm of eIF3 or in eIF3b destabilize binding of TC to the PIC. The $K_D$ of TC (left) or maximal extent of TC recruitment (right) observed for PICs assembled either in the absence of eIF3 (grey bar), or the presence of either WT (blue bar) or variant eIF3 (red bars). Bars and errors represent mean and SEM, respectively (as determined by individual fitting of each experiment), of ≥2 experiments.

The following source data is available for figure 3:

**Source data 1.** Individual measurements of ternary complex binding (dissociation constants and reaction extents) to 43S PICs.

produced temperature-sensitive phenotypes in vivo, we additionally assayed recruitment at 37°C. In these experiments, 43S complexes were formed in the presence of saturating concentrations of TC, eIF1, eIF1A, eIF5, eIF4A, eIF4B, eIF4E•eIF4G, and eIF3; the concentrations of eIF3 and TC were chosen such that they would saturate binding for all variants. Once formed, 43S complexes were incubated with a natural mRNA (*RPL41A*) radio-labeled via a [$^{32}$P]-5'-7-methylguanosine cap. Individual time points were loaded onto a running native polyacrylamide gel, which stops further mRNA binding and resolves free mRNA from mRNA recruited to form 48S complexes (*Figure 4A–B*). The observed 48S complex band is dependent on the presence of an AUG codon in the mRNA (*Mitchell et al., 2010*) (*Figure 4—figure supplement 1A*), and recruitment of capped *RPL41A* mRNA is highly dependent on the presence of both eIF4A and ATP in this in vitro system (*Mitchell et al., 2010*; Yourik, Aitken, Zhou, and Lorsch, unpublished), indicating that it is an active, energy-dependent process. Together, these data suggest that this assay reports on the combined rates of all steps — from initial attachment of mRNA to start-codon recognition — required to form a stable, detectable 48S complex. Hereafter, we will refer to these combined processes simply as 'mRNA recruitment.'

We have previously reported that eIF3 is required for observable recruitment of native mRNA in the reconstituted yeast translation initiation system (*Mitchell et al., 2010*), consistent with its requirement in vivo (*Jivotovskaya et al., 2006*). In the presence of WT eIF3, mRNA recruitment occurs with an apparent rate constant of $0.27 \pm 0.023$ min$^{-1}$ (*Figure 4B*, light blue circles; *Figure 4C*). Mutations throughout eIF3 slow this process. The most dramatic effect occurs in the presence of the a/b/c sub-complex, which slows recruitment more than 10-fold and reduces its endpoint nearly five-fold (*Figure 4B*, orange circles; *Figure 4C*). In contrast, the *Q258R* mutation in eIF3i and the *KLF* mutation in eIF3g have little effect on the kinetics of *RPL41A* mRNA recruitment, indicating that they do not alter regions of these subunits that are critical for rapid loading of mRNA onto the PIC. These mutations confer in vivo phenotypes consistent with defects in either the rate of scanning, as in the case of *Q258R*, or in the processivity of scanning through stable secondary structures, in the case of *KLF*. It is possible that these defects are not rate-limiting for recruitment of the

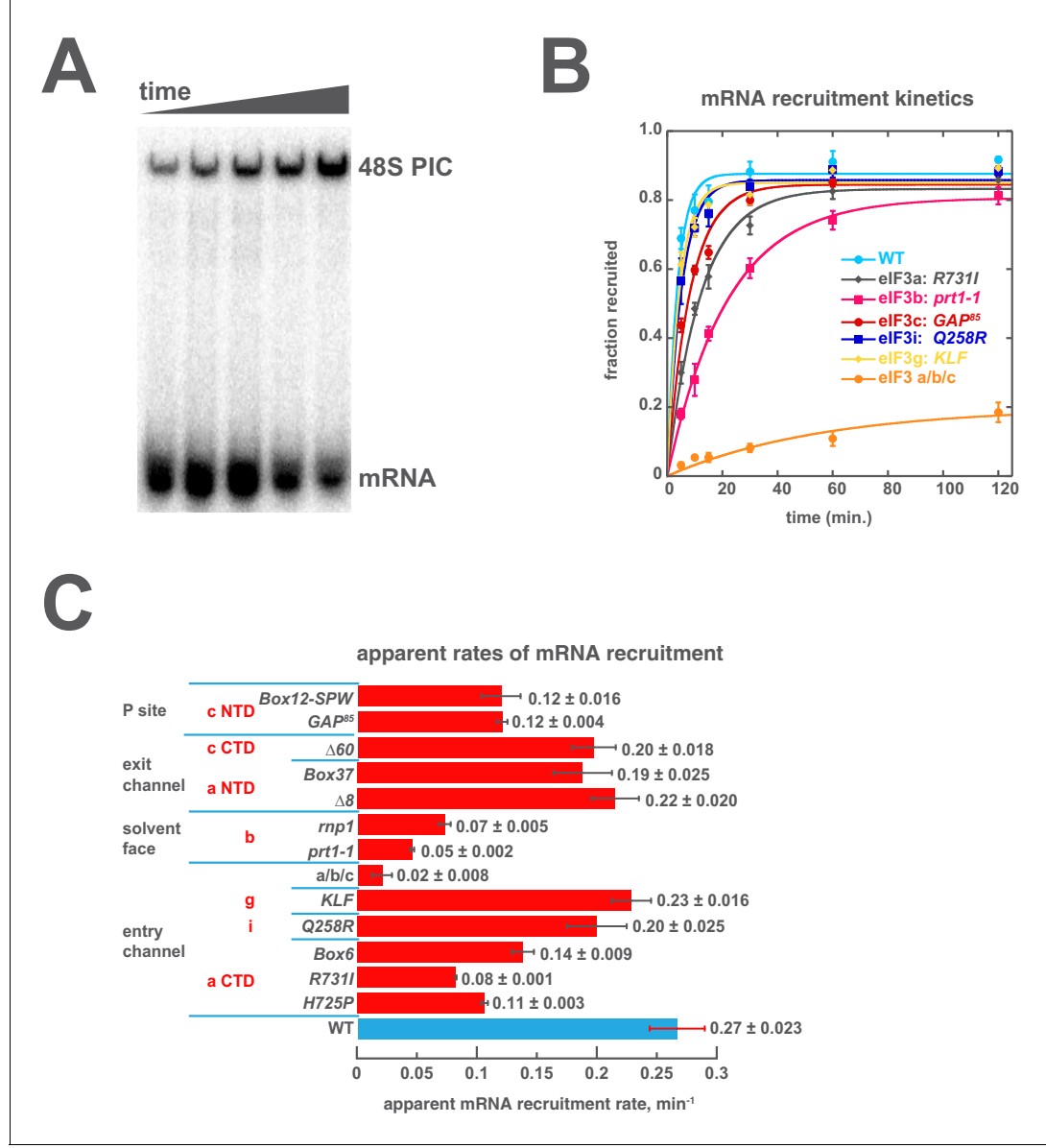

**Figure 4.** Mutations throughout the eIF3 complex compromise its ability to accelerate recruitment of capped native *RPL41A* mRNA to the PIC. (**A**) mRNA recruitment reactions were quenched at appropriate time points on a running native gel to separate free [$^{32}$P]-capped *RPL41A* mRNA from mRNA recruited to form 48S PICs. The bands were quantified to determine the fraction of total mRNA bound at each time point. (**B**) Individual time courses were fit with single-exponential rate equations to determine the observed rate constant for each experiment. (**C**) The observed rates of mRNA recruitment of capped *RPL41A* mRNA measured in the presence of WT (blue bar) and variants of eIF3 (red bars). Bars and errors represent mean values and SEM, respectively (as determined by individual fitting of each experiment), of ≥2 experiments.

The following source data and figure supplements are available for figure 4:

**Source data 1.** Individual observed apparent rates for *RPL41A* mRNA recruitment.

**Figure supplement 1.** The recruitment of unstructured model mRNAs to the PIC depends on the presence of an AUG codon and 40S subunits.

**Figure supplement 2.** Comparison of the kinetics of mRNA recruitment for WT and variant eIF3 at 26°C and 37°C.

*RPL41A* mRNA, which possesses a relatively short (22 nt) 5' UTR lacking highly-structured elements. These defects may become rate-limiting in the recruitment of mRNAs with longer, more-structured 5' UTRs. Nonetheless, the fact that both the extent and rate of recruitment of *RPL41A* mRNA are severely compromised when eIF3 is replaced with the a/b/c sub-complex, together with the previous observation that the loss of eIF3i and eIF3g from the eIF3 complex in vivo critically compromises scanning arrest (*Herrmannová et al., 2012*), is consistent with our interpretation that this assay is sensitive to scanning defects.

The *R731I, H725P,* and *Box6* mutations within the eIF3a CTD slow mRNA recruitment by approximately 2- to 3-fold, further implicating the eIF3 entry channel arm in the mechanism of mRNA recruitment. The eIF3b mutations *prt1-1* and *rnp1* also slow recruitment, by approximately 4- to 5-fold. Aside from the loss of the i and g subunits, these are the strongest effects we observe on the kinetics of recruitment, suggesting that eIF3b also contributes significantly to rapid mRNA recruitment, perhaps by anchoring the entry-channel arm of eIF3 to the solvent face. The eIF3c NTD mutations *GAP[85]*and *Box12-SPW*, thought to be located on the opposite side of the PIC near the P site, also slow mRNA recruitment by ~2 fold. Mutations within the eIF3 exit channel arm, either in the eIF3a NTD (*Δ8* and *Box37*) or the eIF3c CTD (*Δ60*) PCI domains, only slightly reduce the rate of mRNA recruitment; nonetheless, *Box37* significantly slows recruitment at 37°C (*Figure 4—figure supplement 2*), consistent with its temperature-sensitive phenotype in vivo (*Khoshnevis et al., 2014*). Overall, the kinetic defects conferred by mutations throughout eIF3 suggest a requirement for proper collaboration between its different subunits in coordinating rapid mRNA recruitment to the PIC.

One possible explanation for the mRNA recruitment defects we observe in the presence of the eIF3 variants is that mutations in eIF3 affect its ability to collaborate with other factors involved in the mechanism of mRNA recruitment. The eIF4 factors – eIF4A, eIF4B, eIF4E, and eIF4G – have previously been shown to play critical roles during mRNA recruitment both in vivo and in vitro (reviewed in: *Hinnebusch, 2011*; *Jackson et al., 2010*; *Sonenberg and Hinnebusch, 2009*). Our system recapitulates the requirement for these factors in the recruitment of capped natural mRNAs (*Mitchell et al., 2010*) and was previously employed to demonstrate that the absence of eIF4B increases the concentration of eIF4A required for efficient mRNA recruitment (*Walker et al., 2013*), consistent with the genetic interaction between these two factors (*Coppolecchia et al., 1993*; *de la Cruz et al., 1997*). However, varying the concentration of either eIF4A, eIF4B, or eIF4E•eIF4G in mRNA recruitment reactions confirmed that these factors were still saturating for each eIF3 variant under the conditions of our experiments (data not shown). These results suggest that the mRNA recruitment defects we observe do not arise from diminished interaction of the eIF4 factors with the PIC caused by the mutations in eIF3. Moreover, because TC was included at a concentration that saturates its binding in the presence of all variants, we can rule out the possibility that the observed defects are downstream effects of TC binding defects. These in turn imply that eIF3 acts directly to coordinate events during mRNA recruitment, rather than simply helping to recruit TC or the eIF4 factors.

## eIF3 stabilizes the binding of mRNA at the exit channel

The fact that eIF3 is required for mRNA recruitment both in vivo and in vitro and that mutations to both the entry and exit channel arms of eIF3 appear to disrupt mRNA recruitment in vivo (*Chiu et al., 2010*; *Khoshnevis et al., 2014*) suggests that its presence at the mRNA entry and exit channels of the 43S PIC might be central to this role. To investigate the role played by eIF3 at the mRNA entry and exit channels, we next asked how the formation of 48S PICs is affected by the presence or absence of mRNA in each of these channels, and whether eIF3 contributes to the formation of these complexes.

To this end, we replaced the *RPL41A* mRNA in the mRNA recruitment assay with a series of [$^{32}$P]-5'-7-methylguanosine-capped, unstructured model mRNAs, 50 nucleotides in length, in which the location of the AUG codon – located in the 40S P site of the final 48S PIC – dictates the amount of mRNA present in the mRNA entry and exit channels upon start codon recognition (*Figure 5A–C*, left panels; *Figure 5—figure supplement 1*). Based on structural models, the mRNA entry channel can accommodate at least 9 nucleotides 3' of the AUG start codon, and the exit channel is filled with 10 nucleotides 5' of the AUG. As before, we monitored the recruitment of these [$^{32}$P]-5'-7-

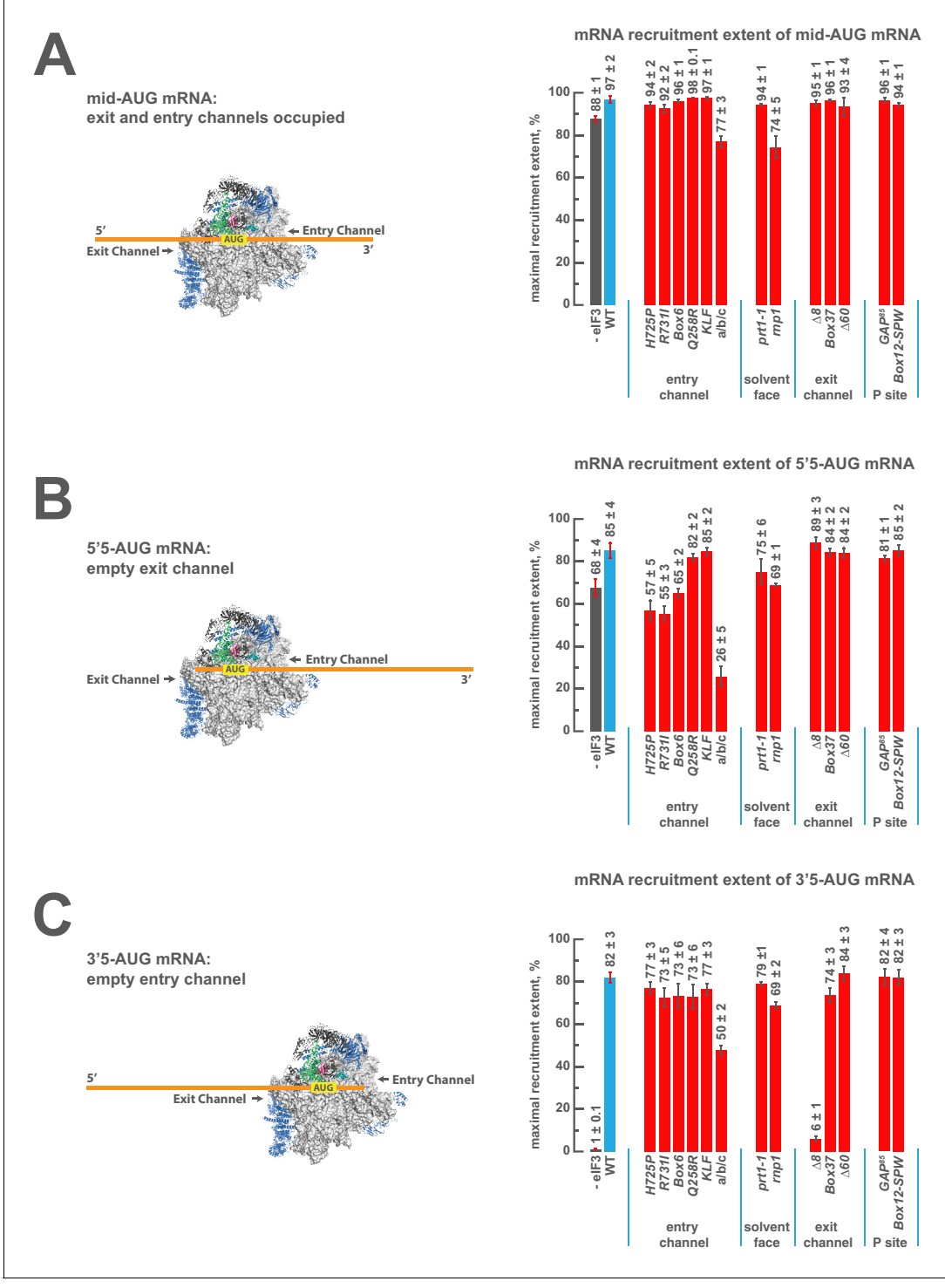

**Figure 5.** eIF3 strongly stabilizes binding of mRNA at the exit channel of the PIC in a manner dependent on the eIF3a NTD. (**A**) The maximal extent of mid-AUG mRNA recruited in the absence of eIF3 (grey bar), or in the presence of either WT (blue bar) or variant eIF3 (red bars). The locations of sequences 5' or 3' of the AUG in the PIC, with AUG in the P site, are shown schematically on the left, indicating that both the mRNA entry and exit channels of the PIC are fully occupied. Bars and errors represent mean and SEM, respectively, of ≥2 experiments. (**B**) The maximal extent of 5'5-AUG mRNA recruited in the absence of eIF3, or in the presence of either WT or variant eIF3. As shown on the left, 5'5 AUG mRNA programs a recruited complex in which only the mRNA entry channel is fully occupied, and thus is sensitized to changes in that channel. (**C**) The maximal extent of 3'5-AUG mRNA recruited in the absence of eIF3, or in the presence of either WT or variant eIF3. The 3'5-AUG programs a recruited complex in which only the mRNA exit channel is occupied (left), and thus is sensitized to changes in that channel.

*Figure 5 continued on next page*

*Figure 5 continued*

The following source data and figure supplements are available for figure 5:

**Source data 1.** Individual measurements of recruitment extent.
**Figure supplement 1.** Short, unstructured model mRNAs with distinct start codon positions.
**Figure supplement 2.** eIF3 accelerates the recruitment of unstructured model mRNAs.
**Figure supplement 3.** The mRNA recruitment extent defects observe for entry- and exit-channel variants of eIF3 are not exacerbated in the absence of the eIF4 factors.

methylguanosine-capped mRNAs in the presence of saturating concentrations of TC, eIF1, eIF1A, eIF5, eIF4A, eIF4B, eIF4E•eIF4G, and eIF3.

In the presence of eIF3, recruitment of a 50 nucleotide mRNA with a centrally positioned AUG codon at nucleotides 24–26 ('mid-AUG') occurs within 15 s, faster than the time resolution of our assay. In the absence of eIF3, however, recruitment proceeds at least two orders of magnitude more slowly, at an apparent rate of $0.02 \pm 0.004$ min$^{-1}$ (*Figure 5—figure supplement 2A*). We observe a similar acceleration of 48S assembly by eIF3 with mRNAs in which the AUG codon occurs 5 nt from the 5' end (5'5-AUG) or 5 nt from the 3' end (3'5-AUG) (*Figure 5—figure supplement 2B–C*).

In contrast, the maximal extent of recruitment of the mid-AUG mRNA is similar in both the presence and absence of eIF3 (*Figure 5A*, grey and blue bars). The extent of mRNA recruitment observed – determined by the endpoint of the reaction defined by multiple time-course experiments – is a function of how much mRNA is recruited in solution and how stably the mRNA is bound within the 48S PIC during gel electrophoresis. Thus, while eIF3 dramatically accelerates the recruitment of the mid-AUG mRNA, it is largely dispensable for stabilizing the binding of the mid-AUG mRNA to this 48S complex, in which both the mRNA entry and exit channels are occupied. Again, mRNA recruitment is strongly dependent on the presence of an AUG codon (*Figure 4—figure supplement 1A–B*), consistent with the interpretation that our assay detects the 48S PIC formed upon start-codon recognition; we similarly observe no recruitment in the absence of 40S subunits (*Figure 4—figure supplement 1C*). Presumably owing to the unstructured nature of the model mRNAs, none requires any of the eIF4 factors for maximum recruitment (*Figure 5—figure supplement 3A–D*, black bars; and data not shown).

We next asked if eIF3 contributes to the stable binding of the 5'5-AUG mRNA, whose recruitment results in a 48S PIC in which the mRNA exit channel is mostly unoccupied, but the entry channel is full (*Figure 5B*, left). The ability to bind this mRNA, as compared to the mid-AUG mRNA, reports on the consequences of disrupting interactions between the 43S PIC and mRNA in the exit channel. In the presence of eIF3, we observe a modest decrease in the extent of recruitment of 5'5-AUG mRNA as compared to mid-AUG mRNA, suggesting that the absence of mRNA in the exit channel slightly destabilizes its binding to the 48S PIC (*Figure 5*, compare blue bars in A and B).

This destabilization resulting from an empty exit channel should make the binding of mRNA to the 48S PIC more dependent on interactions elsewhere within the complex, and thus further sensitize it to disruptions within the entry channel. If eIF3 provides important contacts with mRNA at the entry channel, then its absence should further destabilize mRNA binding in complexes formed on the 5'5-AUG mRNA, in which contacts in the exit channel have already been lost. We observe a moderate decrease in the extent of recruitment in the absence of eIF3 (*Figure 5B*, grey and blue bars), somewhat greater than that observed on the mid-AUG mRNA (*Figure 5A*, grey and blue bars), suggesting that eIF3 makes a moderate contribution to stable binding of mRNA in the entry channel (p<0.01, unpaired, two-tailed t test for -eIF3/+eIF3 ratio on 5'5-AUG mRNA vs. on midAUG mRNA, *Figure 5A and B*, grey and blue bars).

Turning to the 3'5-AUG mRNA, which produces a 48S PIC with the exit channel filled but the entry channel mostly empty, we again observe a modest decrease in recruitment compared to that of mid-AUG mRNA, in the presence of eIF3 (*Figure 5A and C*, blue bars), consistent with the loss of interactions in the entry channel destabilizing mRNA binding to the PIC. Strikingly, however, we observe almost no recruitment of 3'5-AUG mRNA in the absence of eIF3 (*Figure 5C*, grey and blue

bars). Because complexes formed on 3'5-AUG mRNA contain almost no mRNA in the entry channel, they would be expected to be hypersensitive to loss of mRNA interactions in or around the exit channel. Thus, whereas stabilization of mRNA binding in the entry channel does not strongly depend on eIF3 (*Figure 5B*, grey and blue bars), stable binding of mRNA in the exit channel is fully dependent on eIF3. This dependence implicates eIF3 in stabilizing the binding of mRNA either inside or just outside of the exit channel.

## The eIF3a NTD is required for stabilizing the binding of mRNA at the exit channel

We next determined whether mutations in the entry- and exit-channel arms of eIF3 mimic the effects we observe in the absence of eIF3 when either mRNA channel is left empty. As before, all experiments were performed under conditions where binding of TC and each eIF3 variant to the 43S PIC is saturated.

We see no effects on the extent of 48S PIC formation on mid-AUG mRNA – where both channels are occupied – when WT eIF3 is replaced by several eIF3 variants affecting the entry or exit channel arms of the factor, consistent with the observation that WT eIF3 is largely dispensable for the stable binding of mRNA within this complex (*Figure 5A*, blue and red bars). However, the a/b/c sub-complex lacking the i and g subunits confers a modest reduction in endpoint of recruitment slightly greater than that observed even in the absence of the entire eIF3 complex, again highlighting the importance of these subunits to the overall mRNA recruitment process. The *rnp1* variant, which harbors a mutation in the eIF3b RRM located on the 40S solvent face, also confers a similar reduction, perhaps because disruption of this region interferes with the organization of the entry-channel arm to which it connects.

Recruitment of the 5'5-AUG mRNA, which forms a 48S PIC with the exit channel largely empty, produces a different pattern of effects (*Figure 5B*, blue and red bars). Although WT eIF3 only moderately contributes to stable binding of mRNA within this complex, the eIF3a CTD mutations *H725P*, *R731I*, and *Box6* located within the entry channel arm diminish recruitment to essentially the same reduced level observed in the absence of eIF3. This suggests that these mutations impair the moderate contribution of the entry channel arm of eIF3 to stabilizing mRNA binding. Though the eIF3g *KLF* and eIF3i *Q258R* mutations do not destabilize the binding of the 5'5-AUG mRNA, its binding is dramatically reduced in the presence of the a/b/c sub-complex lacking the i and g subunits, even as compared to the absence of eIF3. This is consistent with its modest dominant negative effect on mid-AUG mRNA binding, and suggests that the conformation of the PIC might be altered when these subunits are missing such that the binding of mRNA is destabilized. None of the defects conferred by *H725P*, *R731I*, or the a/b/c subcomplex is exacerbated by the absence of eIF4A, eIF4B, or eIF4E•4G (*Figure 5—figure supplement 3A*), indicating that the eIF3 entry channel arm contributes directly to recruitment of this mRNA, and does not do so simply by binding to these factors. Consistent with the fact that mRNA is already absent from the exit channel in the 5'5-AUG complex, we observe no effects of the eIF3 exit-channel mutations *Δ8*, *Box37*, and *Δ60*.

In contrast, recruitment of the 3'5-AUG mRNA, which produces a complex with an empty entry channel, is dramatically impaired by the *Δ8* truncation. This mutation, which eliminates the first 200 amino acids of the eIF3a NTD PCI domain in the factor's exit-channel arm, mimics the complete absence of eIF3; we observe no recruitment of the 3'5-AUG mRNA in the presence of *Δ8* eIF3 (*Figure 5C*), despite the fact that this mutant factor and TC are both fully bound to the PIC under these conditions (*Figure 2C*, *Figure 3*). Together with the absence of any effect of the *Δ8* mutant on recruitment of the 5'5-AUG mRNA (*Figure 5*, compare Δ8 in B and C), this implicates this region of eIF3a in stabilizing mRNA binding at the exit channel of the 48S PIC. We do not observe any defect in the case of the *Box37* mutation, which maps further into the eIF3a NTD PCI domain, outside the region removed by the *Δ8* truncation. We similarly observe no effect in the presence of the *Δ60* truncation of the eIF3c PCI domain.

Whereas the absence of mRNA in the entry channel sensitizes complexes formed on the 3'5-AUG mRNA to the *Δ8* deletion at the exit channel, it should conversely confer insensitivity to changes at the already vacant entry channel. Accordingly, mutations throughout the eIF3a CTD (*H725P*, *R731I*, *Box6*), eIF3i (*Q258R*), and eIF3g (*KLF*) components of the entry-channel arm do not result in reduced formation of the 3'5-AUG complex (*Figure 5C*). This stands in contrast to the effects of *H725P*, *R731I*, and *Box6* on the 5'5-AUG mRNA (*Figure 5B*), underscoring the deduced defects in entry

channel interactions for these variants. Although we still observe a defect in the presence of the a/b/c sub-complex, this effect is weaker than that observed with the 5'5-AUG complex (*Figure 5*, compare a/b/c in panel B and C). We suggest that the lack of the i and g subunits prevents the factor from promoting the conformation of the PIC needed for optimal mRNA binding.

## The destabilization of mRNA binding caused by eIF3 mutations in either channel can be rescued by sufficient mRNA in the opposite channel

We next set out to determine the minimal length of mRNA in the exit and entry channels required to rescue the defects observed above when those channels are empty. To probe the positions of interactions in the exit channel we measured the extent of recruitment of a model mRNA in which the AUG codon is 11 nucleotides from the 5'-end (5'11-AUG), resulting in a complex with the exit channel mostly occupied, and compared it to the recruitment observed on the 5'5-AUG mRNA, which leaves the exit channel mostly empty and is sensitive to certain entry-channel arm mutations in eIF3. The addition of the six extra nucleotides in the 5'11-AUG mRNA relative to 5'5-AUG mRNA fully or partially rescues the recruitment defects for all of the three eIF3 variants affected by mutations in the eIF3a CTD, leading to recruitment extents nearly as high as those observed with the mid-AUG mRNA that fills the exit channel and extends beyond it (*Figure 6A*). The 5'11-AUG mRNA also partially rescues the defects we observed in the absence of eIF3 and with the a/b/c sub-complex (*Figure 6A*). The ability of 11 nucleotides upstream of the AUG to rescue the destabilizing effects of entry channel mutants is not diminished by the absence of any of the eIF4 factors (eIF4A, eIF4B, eIF4E•eIF4G), suggesting that the interactions restored upon filling the exit channel are with either the 40S subunit itself or with eIF3 at the exit channel pore (*Figure 5—figure supplement 3C*). Consistent with this, the cryo-EM structure of a 48S PIC containing eIF3 shows the −10 nucleotide of the mRNA emerging from the constriction between the 40S head and platform, approximately 15 Å from the position of the eIF3a NTD (*Figure 6B*) (*Llácer et al., 2015*).

We similarly measured the amount of mRNA in the entry channel required to rescue the strong defect in recruitment we observed with the eIF3a Δ8 mutant for 3'5-AUG mRNA that leaves the entry channel largely vacant (*Figure 6C*). The 3'11-AUG mRNA, which programs a recruited complex with 11 nucleotides in the entry channel, partially rescues recruitment in the presence of the Δ8 variant, to an endpoint of 28 ± 1% of total mRNA recruited, but is not sufficient to restore recruitment to the levels observed on the mid-AUG mRNA (*Figure 5A*). Increasing the length of mRNA in the entry channel to 14 and 17 nucleotides progressively rescues the defect, resulting in endpoints of approximately 85 ± 1% and 90 ± 0.3%, respectively (*Figure 6C*). These extents of recruitment approach those observed in the presence of the Δ8 variant for both the 5'5-AUG and mid-AUG mRNAs (*Figure 5A–B*), which both result in a fully-occupied entry channel, suggesting that between 14 and 17 nucleotides of mRNA is sufficient to restore interactions there. The rescue of recruitment we observe with Δ8 and the 5'14-AUG mRNA is not markedly affected by the absence of any eIF4 factor, or eIF5 (*Figure 5—figure supplement 3D*), suggesting that the interactions restored upon filling the entry channel with 14 nucleotides are with either the PIC itself or eIF3. While mRNA is not visible in the entry channel of recent structures of the 48S PIC containing eIF3, a previous structure of a partial 48S complex lacking eIF3 shows the mRNA up to position +12, where it threads past the 40S latch and projects towards a constriction between uS5, uS3, and eS30, and near h16 (*Figure 6D*) (*Hussain et al., 2014*). These 40S components might stabilize mRNA binding by interacting with nucleotides located 14–17 bases downstream of the AUG.

Given the dramatic effect on mRNA recruitment we observe with the eIF3 Δ8 mutant whenever interactions in the entry channel are disrupted, we wondered how these effects might manifest themselves in vivo. Because the cell likely contains no mRNAs resembling the 3'5-AUG mRNA, in which the AUG codon is followed by only 5 nucleotides, we instead reasoned that the Δ8 eIF3 mutant, which is insensitive to the absence of mRNA in the exit channel, might confer a relatively weaker recruitment defect relative to WT cells on mRNAs with extremely short 5'-UTRs, because the majority of mRNA recruitment and scanning on these mRNAs will occur while the exit channel remains empty. In this case, both WT and mutant cells will have the same lack of contacts between mRNA and the PIC in the exit channel. In contrast, the recruitment of mRNAs with longer 5'-UTRs would produce a greater defect for the mutant, as the majority of the scanning process occurs with the exit channel occupied, giving WT PICs an advantage because they are able to make contacts with mRNA in this channel whereas Δ8 mutant cells are not. To investigate this, we quantified the

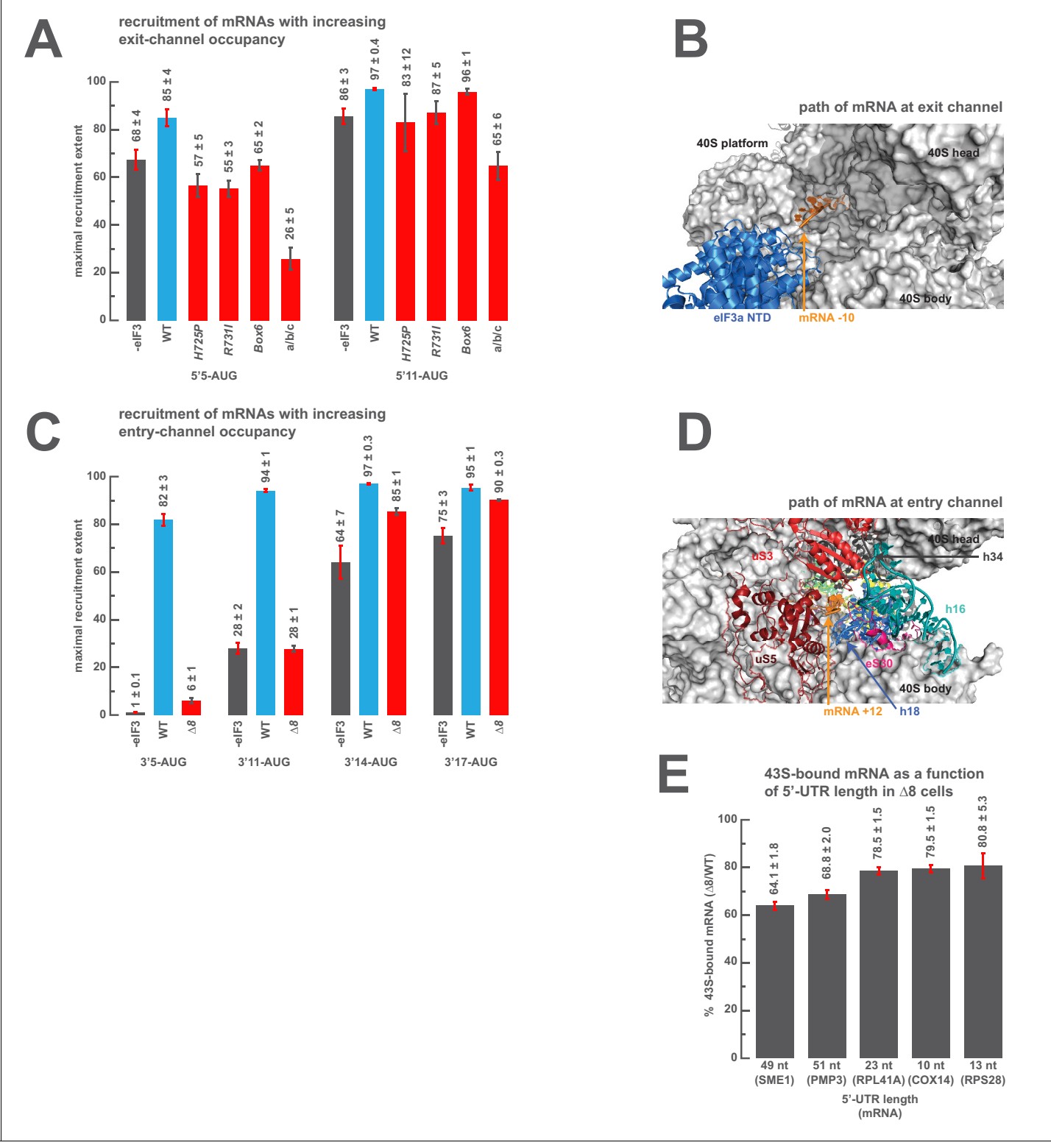

**Figure 6.** The presence of sufficient mRNA in the opposite channel rescues the destabilization of mRNA binding caused by mutations in the entry- or exit-channel arms of eIF3. (**A**) Bars represent the maximal extent of recruitment of mRNAs that program complexes with either 5 (5'5-AUG) or 11 (5'11-AUG) nucleotides in the mRNA exit channel, in the absence of eIF3 (grey bars) or in the presence of either WT eIF3 (blue bars) or variants with mutations near the mRNA entry channel (red bars). Bars and errors represent mean and SEM, respectively, of ≥2 experiments. (**B**) The path of mRNA (orange) at the exit channel of the 40S subunit (grey) in the py48S-closed structure (*Llácer et al., 2015*). mRNA is visualized up to the −10 nucleotide,

*Figure 6 continued on next page*

*Figure 6 continued*

where it emerges from the exit channel pore and is located approximately 15 Å from the eIF3a NTD (blue). (C) Bars represent the maximal extent of recruitment of mRNAs with 5 (3'5-AUG), 11 (3'11-AUG), 14 (3'14-AUG), or 17 (3'17-AUG) nucleotides in the entry channel in the absence of eIF3 (grey bars) or in the presence of either WT eIF3 (blue bars) or the *Δ8* variant (red bars). Bars and errors represent mean and SEM, respectively, of ≥2 experiments. (D) The path of mRNA (orange) at the entry channel of the 40S subunit (grey) in the py48S PIC lacking eIF3 (*Hussain et al., 2014*). mRNA is visualized up to the +12 nucleotide, where it emerges from the 40S latch composed of h34 (charcoal), h18 (blue), and uS3 (red), and projects towards a constriction formed by uS5 (burgundy), uS3 (red), and eS30 (pink), near h16 (teal). (E) The amount of mRNA cross-linked to PICs in *Δ8* cells, as a percentage of mRNA cross-linked to PICs in WT cells, determined for five mRNAs with decreasing 5'-UTR lengths. Bars and errors represent means and SD from two independent biological replicates.

The following source data is available for figure 6:

**Source data 1.** Individual measurements of recruitment extent and of the amount of mRNA associated with 43S PICs in vivo.

amount of mRNA associated with native 43S complexes in vivo, stabilized by formaldehyde cross-linking of living yeast cells, for several mRNAs with different 5' UTR lengths, in strains expressing either WT or the *Δ8* variant of eIF3a. Consistent with our prediction, the *Δ8* mutation reduces the amount of mRNA associated with native 43S PICs for all the mRNAs we tested, but has a greater defect relative to WT cells on mRNAs with longer 5'UTRs (*Figure 6E*).

## Discussion

Despite, and perhaps because of its involvement throughout the translation initiation pathway, the precise molecular roles of eIF3 during the component events of initiation remain unclear. Nonetheless, eIF3 has emerged as a pivotal player in the events that bring the PIC to the mRNA, and ultimately to the start codon. Structural models revealing eIF3 at the 40S solvent face but projecting arms near both the mRNA entry and exit channels, as well as genetic and biochemical evidence consistent with eIF3 operating in these regions, suggest the compelling possibility that eIF3 mediates events such as mRNA recruitment and scanning via these appendages. Our data indicate that while regions throughout eIF3 collaborate to stabilize its binding to the 43S PIC and directly promote mRNA recruitment, the mRNA entry- and exit-channel arms of eIF3 play distinct roles within the PIC. The eIF3 exit-channel arm — and in particular the eIF3a NTD — is required to stabilize the binding of mRNA at the exit channel within the 48S PIC (*Figure 7*). In contrast, the entry-channel arm helps to stabilize mRNA binding in the entry channel but also participates in stabilizing TC binding to the PIC and in accelerating mRNA recruitment (*Figure 7*).

### eIF3 binds the PIC via multiple interactions and contributes directly to mRNA recruitment

Our results support the idea that the binding of eIF3 to the PIC is mediated by multiple interactions. Mutations in both the eIF3a CTD and NTD, eIF3b, eIF3c, as well as the simultaneous absence of eIF3i and eIF3g weaken the affinity of eIF3 for the PIC. This is consistent with the multiple contacts observed between eIF3 and the PIC in recent structures (*Aylett et al., 2015*; *Georges, des et al., 2015*; *Llácer et al., 2015*), and with the diverse set of interactions between eIF3 subunits and components of the PIC identified in genetic and biochemical studies (*Asano et al., 2000*; *Chiu et al., 2010*; *Pisarev et al., 2008*; *Szamecz et al., 2008*; *Valásek, 2012*; *Valásek et al., 2003*). Notably, we observe that none of the eIF3 variants displays a severe defect in PIC binding, despite the fact that several directly affect regions that appear to interact with the 40S subunit in the py48S-closed structure or target previously identified interactions. This suggests that because the factor makes multiple contacts with the PIC, no single interaction is itself essential for binding.

Our results further support a direct role for eIF3 in promoting mRNA recruitment. None of the eIF3 variants we tested that significantly slow recruitment of *RPL41A* mRNA do so as a result of the concentrations of TC, eIF4A, eIF4B, or eIF4E•eIF4G becoming limiting under the conditions of our assay. While this does not preclude an interaction between eIF3 and these other factors, it does suggest that eIF3 contributes directly to mRNA recruitment. This is consistent with our previous finding that eIF3 is essential, whereas eIF4G and eIF4B are only rate-enhancing, for recruitment of *RPL41A*

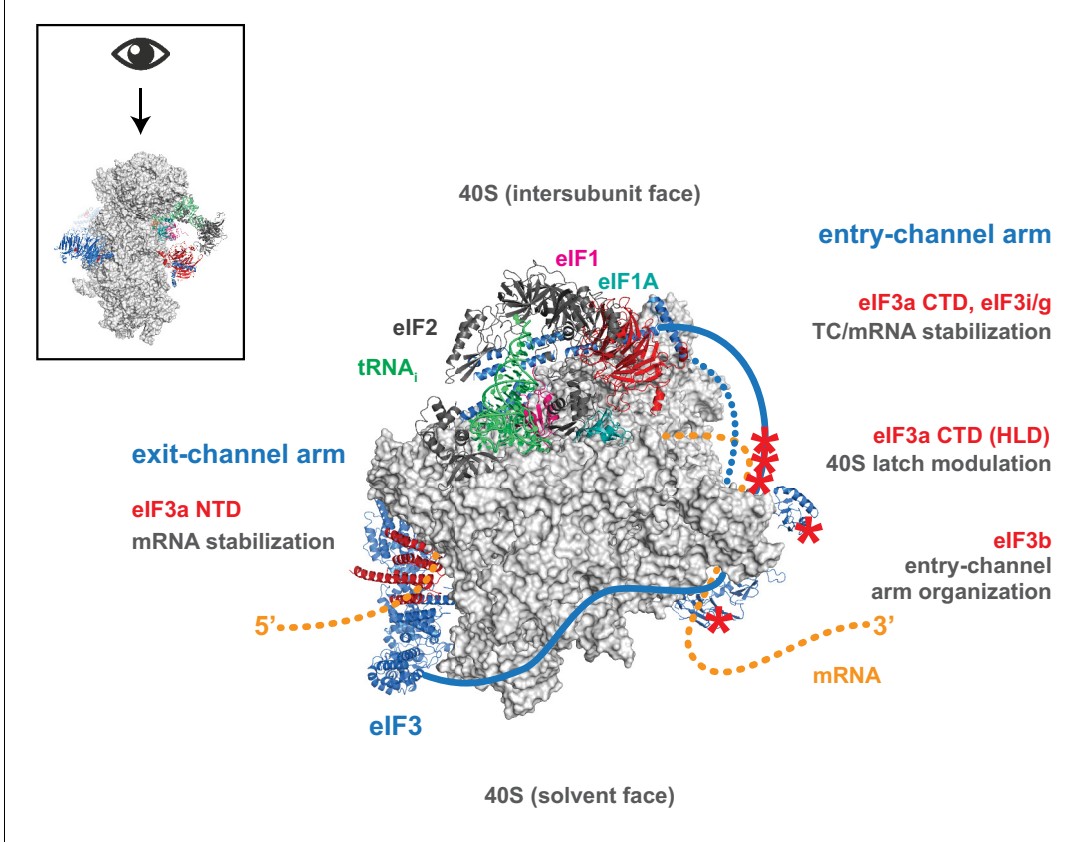

**Figure 7.** Distinct roles for the mRNA entry- and exit-channel arms of eIF3. A proposed model for the roles of the eIF3 entry- and exit-channel arms during translation initiation. The py48S-closed complex is shown from above (boxed schematic), looking down at the head of the 40S (light grey), with the solvent face at the bottom, and the intersubunit face at the top with Met-tRNAi (green), eIF2 (dark grey), eIF1 (pink), and eIF1A (teal) bound to the decoding center. eIF3 is shown in blue, with areas of interest highlighted in red or denoted with red asterisks; resolved regions of eIF3 are depicted as ribbons, with unresolved regions cartooned as solid or dashed lines. The mRNA is cartooned in orange, entering the PIC though the entry channel at right, and exiting at the pore near the platform at left. Our results suggest that the eIF3 exit-channel arm, and specifically the eIF3a NTD, is critical for stabilizing mRNA binding to the PIC at the exit channel, whereas the eIF3a CTD (unresolved regions appear as solid blue line) and eIF3i/g enhance the stability of mRNA interactions at the entry-channel. The entry-channel arm and its attachment to the solvent face via eIF3b (unresolved regions appear as dashed blue line) is also important for stabilizing TC binding to the PIC and promoting steps that control the kinetics of mRNA recruitment, perhaps as a function of modulating the 40S latch.

mRNA in vitro (*Mitchell et al., 2010*), and with the observation that depletion of eIF3 from yeast essentially abolishes recruitment of the *RPL41A* mRNA in vivo (*Jivotovskaya et al., 2006*).

## The eIF3 entry-channel arm stabilizes TC binding to the PIC and is important for recruitment of native mRNA

Our results point to a role for the eIF3 entry channel arm — comprising the eIF3a CTD, eIF3i, eIF3g, and eIF3b — in stabilizing both eIF3 and TC binding to the PIC, and making important contributions to the mechanism of mRNA recruitment. In our experiments, WT eIF3 binds tightly to both 43S and 48S PICs and strongly stabilizes TC binding to the PIC in the absence of mRNA. The strong stabilization of TC binding we observe is consistent with genetic evidence that eIF3 promotes TC recruitment in vivo (*Asano et al., 2000*; *Phan et al., 2001*; *Valasek et al., 2004*; *Valášek et al., 2002*).

In contrast, the a/b/c sub-complex lacking eIF3i and eIF3g displays weakened affinity for both 43S and 48S PICs (*Figure 2C*) and diminishes the affinity of the PIC for TC relative to that observed with WT eIF3 (*Figure 3*). In a structure of yeast eIF3 bound to the 40S•eIF1•eIF1A complex (*Aylett et al., 2015*), eIF3i and eIF3g are found distant from TC on the solvent-face side of the PIC above the eIF3b β-propeller domain, and appear to interact with eIF3b but not with the 40S subunit.

This position agrees with the placement of eIF3i in the structure of mammalian eIF3 bound to a complex containing DHX29 (*Georges, des et al., 2015*), though eIF3g was not resolved. In the yeast py48S-closed PIC (*Llácer et al., 2015*), however, these subunits are found at the intersubunit face below the A site, together with the C-terminus of eIF3b; both eIF3i and eIF3g appear to interact with the 40S and eIF3i appears to contact the eIF2γ subunit of TC, consistent with our observation that the absence of eIF3i and eIF3g weakens the binding of both TC and eIF3 to the PIC. Further evidence of a role for eIF3i and eIF3g in contributing to PIC binding by eIF3 is provided by the in vivo effects of the eIF3i *DDKK* mutation, which results in the dissociation of both of these subunits from the eIF3 complex and reduces the amount of a/b/c sub-complex bound to PICs in vivo (*Herrmannová et al., 2012*).

Most strikingly, we observe that the a/b/c sub-complex lacking eIF3i and eIF3g is essentially unable to promote the recruitment of capped *RPL41A* mRNA to the PIC (*Figure 4B–C*). This stands in contrast to a previous study where the affinity-purified a/b/c sub-complex rescued mRNA recruitment in heat-treated extracts from *prt1-1* cells (*Phan et al., 2001*). However, these extracts were not deprived of WT eIF3, perhaps enabling reconstitution of the complex with WT eIF3i and g subunits. In fact, the simultaneous loss of eIF3i and eIF3g from the eIF3 complex in *DDKK* cells produces phenotypes consistent with significant defects in scanning and start-codon recognition (*Herrmannová et al., 2012*). Because mRNA recruitment in our assay depends on the presence of an AUG codon, the kinetics we observe may report not only on initial mRNA attachment, but also on the subsequent events of scanning and start-codon recognition, suggesting an inability to efficiently scan and recognize the AUG codon in the absence of eIF3i and g prevents efficient *RPL41A* recruitment. In light of the severe mRNA recruitment defects we observe in the simultaneous absence of eIF3i and eIF3g, it is notable that neither the *Q258R* eIF3i mutation nor the *KLF* eIF3g mutation confer significant defects in mRNA recruitment. One possible explanation for these apparently opposed observations is that these mutations, which appear to affect either the rate or processivity of scanning, confer defects that are not rate limiting on the *RPL41A* mRNA employed in our assays, whose 5′ UTR is relatively short. It will be interesting to see the effects of these and other mutations that affect the efficiency of scanning by the PIC on the in vitro recruitment of natural mRNAs with longer, more structured 5′ UTRs.

Both proposed locations for eIF3i and eIF3g place them adjacent to the mRNA entry channel, either on the solvent or intersubunit face of the PIC (*Cuchalova et al., 2010*; *Aylett et al., 2015*; *Llácer et al., 2015*). Together with our observation that these subunits are critical for *RPL41A* mRNA recruitment in vitro and scanning and start-codon recognition in vivo (*Cuchalova et al., 2010*; *Herrmannová et al., 2012*), the observation that these subunits appear at the intersubunit face in structures containing mRNA, but at the solvent face in structures lacking mRNA further suggests that the mRNA entry channel arm of eIF3 might undergo a large-scale conformational change upon initial binding of the mRNA which could facilitate processive scanning and its arrest.

The *H725P and R731I* mutations within the eIF3a CTD also interfere with eIF3 binding to the PIC (*Figure 2C*) and, together with the *Box6* mutation, significantly slow the kinetics of *RPL41A* mRNA recruitment (*Figure 4C*). In contrast to the absence of the eIF3i and eIF3g subunits, these mutations do not appear to reduce the affinity of the PIC for TC (*Figure 3*), consistent with the observation that the interaction between eIF3a and eIF2 occurs in a region downstream of these mutations (*Valášek et al., 2002*). Though structures of eIF3 bound to various states of the PIC have yet to completely resolve the eIF3a CTD, previous work has established interactions between it and the eIF3b RRM (*Valasek et al., 2001b*), components of the 40S latch (*Chiu et al., 2010*; *Valášek et al., 2003*), and TC (*Valášek et al., 2002*), locating it near the entry channel and even projecting towards the intersubunit face. These observations enabled its proposed placement in the yeast py48S-closed structure, where it spans the intersubunit face, nearly encircling the PIC (*Llácer et al., 2015*). The binding defects we observe may result from disruptions to interactions between the eIF3a CTD and 40S latch; these disruptions might also affect the opening and closing of the latch, explaining the recruitment defects conferred by eIF3a CTD mutations. Consistent with this, all three eIF3a CTD mutations reduce the amount of *RPL41A* mRNA associated with PICs in vivo (*Chiu et al., 2010*). In particular, both the *H725P and R731I* mutations impair mRNA recruitment without affecting upstream events in vivo, and appear to affect the equilibrium between the open and closed conformations of the PIC.

We further observe a similar profile of effects in the presence of mutations to eIF3b. The *prt1-1* and *rnp1* mutations in the eIF3b β-propeller and RRM domains, respectively, disrupt both eIF3 and TC binding to the PIC (*Figure 2C* and *Figure 3*), and slow *RPL41A* mRNA recruitment (*Figure 4C*). In fact, these mutations confer the strongest kinetic effects we observe on native mRNA recruitment. Consistent with these results, both mutations produce in vivo phenotypes suggesting significant defects in scanning and start-codon recognition (*Nielsen et al., 2004*, *2006*) and both have also been shown to reduce the amount of eIF3 associated with PICs either in cells or, in the case of *prt1-1*, in extracts. In several structures of eIF3 bound to the PIC, the eIF3b β-propeller domain is seen interacting with the 40S solvent face, below h16 (*Figure 7*) (*Aylett et al., 2015*; *Georges, des et al., 2015*; *Llácer et al., 2015*), consistent with its known interaction with uS4 (*Liu et al., 2014*). The eIF3b C-terminus, which appears together with eIF3i and eIF3g at the intersubunit face in the yeast py48S-closed structure, has been shown to mediate interactions with eIF3i (*Asano et al., 1998*; *Herrmannová et al., 2012*). Together with the our results and the previous observation that the eIF3b RRM interacts with the eIF3a CTD (*Valasek et al., 2001b*), these raise the possibility that eIF3b contributes to the working of the eIF3 entry-channel arm by anchoring it to the 40S solvent face and, via its RRM and C-terminal regions, organizing its constituent components. The *prt1-1* and *rnp1* mutations might disrupt this mechanistic nexus, explaining the diverse set of defects they confer.

Another possibility is that eIF3b contributes more directly to mRNA recruitment, either from its position below the entrance to the mRNA entry channel on the solvent face as observed in the majority of structures, or by relocating to the intersubunit face. A recent report located mammalian eIF3i to a distinct location on the intersubunit face and suggested that the eIF3b β-propeller instead interacts with TC in the yeast 48S PIC (*Simonetti et al., 2016*). While this model would not explain the TC binding defects we observe in the absence of eIF3i and eIF3g, it would nonetheless still be consistent with the defects in TC affinity that we observe in the presence of eIF3b mutants and with a role for the eIF3 entry channel arm in stabilizing TC binding to the PIC.

## The eIF3 exit-channel arm and the eIF3c NTD contribute to PIC binding and mRNA recruitment

Both the *Δ8* truncation and the *Box37* substitution in the eIF3a NTD diminish the affinity of eIF3 for 43S and 48S PICs. Consistent with this, structures of both yeast and mammalian eIF3 bound to various states of the PIC reveal the PCI domain in the eIF3a NTD, and specifically residues within the region truncated in the *Δ8 mutant*, as well as the eIF3c C-terminal PCI domain, contacting the 40S on its solvent face, below the platform (*Aylett et al., 2015*; *Georges, des et al., 2015*; *Llácer et al., 2015*). Moreover, the eIF3a NTD has been shown to interact with uS2 (*Valášek et al., 2003*; *Kouba et al., 2012b*) which is located in this vicinity, and the *Δ8* truncation reduces the amount of eIF3 associated with PICs in vivo (*Szamecz et al., 2008*). In contrast, the *Box37* mutation, which confers a Ts⁻ phenotype in vivo, appears to primarily affect mRNA recruitment in cells (*Khoshnevis et al., 2014*). In fact, we observe a significant defect in *RPL41A* mRNA recruitment at 37°C in the presence of *Box37* eIF3 (*Figure 4—figure supplement 2*). Based on structures, the *Box37* mutation does not appear to occur on the surface of the eIF3a PCI domain that faces the mRNA exit channel pore (*Llácer et al., 2015*), and thus may exert its effects indirectly by disrupting this domain, as suggested previously (*Khoshnevis et al., 2014*).

Elsewhere, the *Box12-SPW* and *GAP⁸⁵* mutations to the eIF3c NTD also slow native mRNA recruitment. Both mutations are in a region that appears to interact with eIF1 and both affect the amount of eIF1 associated with PICs and the accuracy of start-codon recognition in vivo (*Karásková et al., 2012*). The defects we observe are consistent with the observation that mRNA recruitment in our assay depends on codon-recognition (*Figure 4—figure supplement 1A–B*). The *Box12-SPW* mutation additionally weakens the affinity of eIF3 for the PIC. While this region has not been well-resolved in structural studies, regions of the eIF3c NTD were modeled on the 40S intersubunit face below the platform in the yeast py48S-closed structure, suggesting that this putative interaction might contribute to stabilizing eIF3 binding to the PIC (*Llácer et al., 2015*). Another possibility is that both mutations exert their effects on PIC binding via their effects on the eIF3:eIF1 interaction.

## The eIF3a NTD is critical for stabilizing mRNA binding at the exit channel

To investigate the molecular role of eIF3 in stabilizing mRNA binding to the PIC at both the entry and exit channels, we followed the recruitment of a series of capped model mRNAs in which the placement of the AUG codon enables us to control the amount of mRNA in either channel. These experiments reveal that, while eIF3 is dispensable for stabilizing the binding of mRNA when both the entry and exit channels are occupied, it is essential for doing so when the entry channel is empty and the complex thus depends on interactions with mRNA in the exit channel. Previous work demonstrated that mammalian eIF3 can be cross-linked to mRNA at positions −8 thru −17 (relative to the AUG at +1 to +3), indicating its proximity to mRNA nucleotides just inside the entry channel (−8 to −10) and protruding from the entry channel pore (−11 to −17) (*Pisarev et al., 2008*). Consistent with this, our results indicate that eIF3 stabilizes the binding of mRNA to the PIC at this location.

In fact, our results localize this role to the eIF3a NTD; the Δ8 truncation mimics the absence of eIF3 in its inability to stabilize the binding of mRNA to 48S PICs in which the entry channel is mostly empty (*Figure 5C*). This is consistent with the observation that the Δ8 truncation impairs reinitiation on *GCN4* mRNA following translation of uORF1 and uORF2, an effect attributable to the loss of interaction between the eIF3a NTD and nucleotides 5′ of each uORF stop codon that otherwise stabilize post-termination 40S subunits and allow them to resume scanning in the presence of WT eIF3 (*Gunišová et al., 2016*; *Munzarová et al., 2011*; *Szamecz et al., 2008*). This role of the eIF3a NTD is also consistent with its placement immediately adjacent to the exit-channel pore in several structures of eIF3 bound to the 40S, in both yeast and mammals (*Aylett et al., 2015*; *Georges, des et al., 2015*; *Llácer et al., 2015*). In the cryo-EM structure of a yeast 48S PIC containing eIF3, the −10 nucleotide of the mRNA is visible as it begins to emerge from the exit channel at the solvent face (*Figure 6B*); we showed that the presence of 11 nucleotides within the exit channel is sufficient to rescue the destabilization caused by mutations affecting the entry channel arm of eIF3 (*Figure 6A*). This effect suggests that the eIF3a NTD might serve to extend the mRNA exit channel behind the 40S platform, interacting directly with mRNA as it emanates from the exit-channel pore. Another possibility is that the eIF3a NTD influences the conformation of the exit channel, promoting stable interaction between the mRNA and components of the 40S within the channel. Thus, the established stimulatory effects on translation initiation conferred by a minimum 5′UTR length and favorable sequence context just upstream of the start codon might also depend on proper interactions of 5′UTR nucleotides with the 40S exit channel, and hence, could be influenced by the eIF3a-NTD.

In contrast to its essential role in the binding of mRNA at the exit channel, eIF3 more modestly stabilizes mRNA binding at the entry channel. And yet, the absence of eIF3, or the presence of eIF3 variants with eIF3a-CTD mutations *H725P, R731I,* or *Box6*, all reduce binding of a model mRNA in which the exit channel is empty but the entry channel is filled (*Figure 5B*); these binding defects are largely rescued by increasing the amount of mRNA upstream of the AUG codon to fill the exit channel completely (*Figure 6A*). These results implicate the eIF3a CTD in stabilizing mRNA interactions at the entry channel of the PIC in a manner that is redundant with mRNA interactions at the entry channel involving the eIF3a NTD. Consistent with this, the presence of increasing amounts of mRNA in the entry channel progressively rescues the destabilization of mRNA binding to the PIC conferred by the Δ8 variant, which is defective for interactions at the exit channel (*Figure 6C*). These observations might reflect the importance of a constriction of the mRNA entry channel formed by a movement of the 40S head that appears to be triggered by start codon recognition (*Llácer et al., 2015*; *Zhang et al., 2015*). This head movement also closes the latch on the entry channel, a feature composed of interactions between 18S rRNA helices 18 and 34 and ribosomal proteins uS3/uS5, which might contribute to scanning arrest upon start-codon recognition (*Llácer et al., 2015*). One explanation for our finding that full rescue of the Δ8 exit channel defect requires filling the entry channel with ~14–17 nucleotides is that this represents the minimum length of mRNA required to fully engage the 40S components of the entry channel in its restricted conformation within the closed PIC and to also pass through the constriction created by the closed latch and interact with elements beyond it, which could include the eIF3a CTD. In fact, the structure of a partial yeast 48S complex lacking eIF3 (py48S) shows that 9 nt 3′ of the AUG codon (nucleotide + 12) is long enough to pass beyond h18/h34, but not past uS3/uS5 (*Figure 6D*) (*Hussain et al., 2014*). It is notable that, with the

exception of h34, every 40S component of the latch has been implicated in interactions with eIF3 (*Chiu et al., 2010*; *Valášek et al., 2003*).

Interestingly, the a/b/c sub-complex lacking the i and g subunits produces a stronger defect in recruitment of the model mRNA lacking contacts in the exit channel compared to the absence of eIF3 itself, perhaps because the absence of the i and g subunits alters the conformation of this entry channel arm, inducing a conformation of the PIC or of eIF3 itself that inhibits mRNA binding to the PIC. This might occur as a result of disruptions to the proposed rearrangement of the entry-channel arm upon mRNA binding to the PIC. In the end, the role of the eIF3 entry channel arm appears distinct from its counterpart at the exit channel (*Figure 7*). Whereas it contributes more subtly to stabilizing mRNA binding to the PIC, it contributes indirectly to mRNA recruitment by stabilizing TC binding and, as evidenced by its importance in the recruitment of natural mRNA, participates in events that control the rate of mRNA recruitment or scanning, perhaps by responding to the presence of mRNA within the PIC and by modulating the conformation of the 40S latch.

The work presented here provides greater understanding of how the individual subunits of eIF3 contribute to its interaction with the PIC, and to the component events of the initiation pathway. In particular, our results dissect the distinct roles played by eIF3 at the mRNA entry and exit channels, and how these contribute to the events of mRNA recruitment. Future work is required to better understand the molecular details of these contributions, particularly in the entry channel, where eIF3 appears to play a more subtle mechanistic role, perhaps collaborating with the 40S itself to mediate interaction with the mRNA. It will also be interesting to determine how, if at all, yeast eIF3 collaborates with the other mRNA recruitment factors to drive this event.

## Materials and methods

### Purification and preparation of reagents

40S ribosomal subunits and individual initiation factors were purified as previously described (*Acker et al., 2007*; *Mitchell et al., 2010*). eIF3 variants were purified from corresponding strains using the same method as for the wild-type factor, with the modification that cells were grown in the appropriate selective media throughout. Initiator tRNA$_i^{Met}$ was transcribed in vitro using His$_6$-tagged T7 RNA polymerase and subsequently methionylated with purified N-terminal His$_6$-tagged *E. coli* methionyl-tRNA synthetase as previously described (*Walker and Fredrick, 2008*), using either unlabeled methionine or [$^{35}$S]-methionine (PerkinElmer, Waltham, MA). All mRNAs were transcribed in vitro and purified as previously described (*Mitchell et al., 2010*). mRNAs were capped using the vaccinia virus D1/D12 capping enzyme as previously described (*Mitchell et al., 2010*). *RPL41A* mRNA was capped at a concentration of 5 μM, in the presence 50 mM Tris-HCl (pH 7.8), 1.25 mM MgCl$_2$, 6 mM KCl, 2.5 mM DTT, 50 μM GTP, 100 μM S-adenosylmethionine, 2 μCi/μL α-[$^{32}$P]-GTP [Perkin Elmer], 125 nM capping enzyme, 1 U/μL RiboLock RNase Inhibitor [Invitrogen]). Model mRNAs were capped at a concentration of 50 μM under similar conditions, except the concentration of cold GTP was increased to 100 μM to ensure it remained in excess of the mRNA. Capping reactions were incubated at 37°C for 90 min, and mRNAs were subsequently purified using the RNeasy mini kit (Qiagen). Model mRNAs were purified via a modified RNeasy mini kit protocol: 10 μL capping reactions were brought to 30 μL total volume using RNase-free water, followed by addition of 320 μL RLT buffer (Qiagen) and mixing by vortexing. Upon mixing, 525 μL of 100% ethanol was added, and the resultant solution was mixed by careful pipetting. Half of this mixture was loaded on to an RNeasy spin column and centrifuged at ≥8000 ×g for 15 s, after which the flowthrough was discarded, and the remaining mixture was loaded onto the column and centrifuged, and the flowthrough again discarded. The column was then sequentially washed once with 700 μL of RWT buffer (Qiagen) and twice with 500 μL buffer RPE (Qiagen), with each wash followed by centrifugation for 15 s at ≥8000 ×g and disposal of the flowthrough. Upon the final wash, centrifugation at ≥8000 ×g was extended to 60 s and performed twice, with the flowthrough discarded each time, to ensure complete drying of the column. Purified RNA was then eluted by addition of 30 μL RNase-free water to the column and centrifugation for 60 s at ≥8000 ×g.

## Biochemical assays

The affinity of WT and variant eIF3 for 43S and 48S PICs was determined using a gel-shift assay monitoring [$^{35}$S]-Met-tRNA$_i$$^{Met}$ (*Acker et al., 2007*; *Mitchell et al., 2010*). 43S complexes (in the absence of mRNA) were formed in the presence of 1 μM eIF1, 1 μM eIF1A, 200 nM eIF2, 1 mM GDPNP•Mg$^{2+}$, 2 nM [$^{35}$S]-Met-tRNA$_i$$^{Met}$, and 30 nM 40S subunits in 1X Recon buffer (30 mM HEPES-KOH pH 7.4, 100 mM KOAc pH 7.6, 3 mM Mg(OAc)$_2$, 2 mM DTT). The concentration of WT or variant eIF3 was titrated between 30 nM and 500 nM. 48S complexes (including mRNA) were formed in the presence of 1 μM uncapped AUG-containing model mRNA, 1 μM eIF1, 1 μM eIF1A, 200 nM eIF2, 1 mM GDPNP•Mg$^{2+}$, 2 nM [$^{35}$S]-Met-tRNA$_i$$^{Met}$, and 20 nM 40S subunits. The concentration of either WT or variant eIF3 was titrated between 30 nM and 500 nM for both. 10 μL reactions were incubated at 26°C for 2 hr, after which they were mixed with 2 μL loading buffer (0.05% xylene cyanol, 0.05% bromophenol blue, 50% sucrose in 1X Recon buffer) and resolved on a 4% polyacrylamide (37.5:1 acrylamide:bisacrylamide) gel prepared in 1X THEM (34 mM Tris Base, 57 mM HEPES, 0.1 mM EDTA, 2.5 mM MgCl$_2$) gel run at 25 W for 45 min using 1X THEM as the running buffer. Upon completion, gels were placed on Whatman paper, covered with plastic wrap, and exposed for at least 15 h to a phosphor screen inside a cassette wrapped in plastic wrap at −20°C. Cooled cassettes were removed from refrigeration at −20°C and allowed to warm at room temperature for approximately 10 min prior to removal of the plastic wrap and scanning of the phosphor screen using a Typhoon FLA 9500 imager (GE Life Sciences). The fraction of eIF3-bound PICs was determined from the signal of free and eIF3-bound PIC bands. Data were modeled with both Langmuir and Hill-binding isotherms to determine the apparent K$_D$.

The affinity of 43S PICs for TC was determined using a previously-described gel-shift assay monitoring [$^{35}$S]-Met-tRNA$_i$$^{Met}$ (*Acker et al., 2007*; *Kolitz et al., 2009*). Complexes were formed in the presence of 1 μM eIF1, 1 μM eIF1A, 200 nM eIF2, 1 mM GDPNP, 1 nM [$^{35}$S]-Met-tRNA$_i$$^{Met}$, and 500 nM eIF3 in 1X Recon buffer (30 mM HEPES-KOH pH 7.4, 100 mM KOAc pH 7.6, 3 mM Mg(OAc)$_2$, 2 mM DTT). The concentration of 40S subunits was titrated between 10 nM and 320 nM. Reactions were incubated at 26°C for 2 hr and subsequently resolved on a 4% native THEM gel run at 25 W for 45 min. Gels were exposed and quantified as above, with the fraction of TC binding quantified as the ratio of signal from bands representing eIF3-bound PICs and free TC. Data were modeled with a Langmuir isotherm to determine the apparent K$_D$.

The kinetics and extent of mRNA recruitment to reconstituted 43S PICs were determined using a previously-described gel-shift assay monitoring [$^{32}$P]-capped mRNA (*Mitchell et al., 2010*). PICs were assembled in the presence of 1 μM eIF1, 1 μM eIF1A, 300 nM eIF2, 200 nM Met-tRNA$_i$$^{Met}$, 400 nM eIF3, 2 μM eIF4A, 300 nM eIF4B, 50 nM eIF4E•eIFG, 300 nM eIF5, and 30 nM 40S subunits in 1X Recon buffer (30 mM HEPES-KOH pH 7.4, 100 mM KOAc pH 7.6, 3 mM Mg(OAc)$_2$, 2 mM DTT) and incubated 10 min at 26°C. Reactions were initiated by the simultaneous addition of ATP•Mg$^{2+}$ and the appropriate [$^{32}$P]-capped mRNA to final concentrations of 2 mM and 15 nM, respectively. For kinetic measurements, 4 μL aliquots were removed at appropriate time points, added to 1 μl of loading buffer and quenched by loading on a 4% native THEM gel running at 200 V. Gels were exposed and quantified as above, with the fraction of mRNA recruitment quantified as the fraction of signal in the band representing 48S PICs per lane. Data were fit with single-exponential rate equations to obtain the observed rate constant of mRNA recruitment. To measure the maximal extent of recruitment at completion, reactions were similarly resolved on a 4% native THEM gel upon incubation for 2 hr at 26°C, at which point all reactions had proceeded to completion as judged by prior kinetic experiments. For experiments containing *RPL41A* mRNA, gels were run for 1 hr at 200 V. For experiments containing 50 nt model mRNAs, gels were run 45 min at 200 V.

## In vivo assays

The analysis of native 48S PICs was performed as previously described (*Chiu et al., 2010*; *Khoshnevis et al., 2014*). Cells were cultured at 30°C to OD600$_{600}$ = 1 and cross-linked with 1% formaldehyde prior to being harvested. Whole Cell Extracts were prepared in breaking buffer (20 mM Tris-HCl, pH 7.5, 50 mM KCl, 10 mM MgCl$_2$, 1 mM dithiothreitol [DTT], 5 mM NaF, 1 mM phenylmethylsulfonyl fluoride [PMSF], 1× Complete Protease Inhibitor Mix tablets without EDTA [Roche]), and 25 A260 units were separated on a 7.5% to 30% sucrose gradient by centrifugation at 41,000 rpm for 5 hr in a SW41Ti rotor. Total RNA was isolated from 500 μL of each of 13 gradient

fractions by hot-phenol extraction, and resuspended in 26 µL of diethyl pyrocarbonate-treated $H_2O$. Prior to RNA isolation, 3 µL of TATAA Universal RNA Spike I (Tataa Biocenter) was added to each fraction as a normalization control. 3 µL of total RNA were subjected to reverse transcription using the High Capacity cDNA Reverse Transcription Kit (Applied Biosystems). Aliquots of cDNA were diluted six-fold, and qPCR amplifications were performed on 2 µL of diluted cDNA in 10 µL reaction mixtures prepared with the 5x HOT FIREPol EvaGreen qPCR Supermix (Solis Biodyne) and primers for RPL41A (0.3 µM), 18S rRNA, PMP3, SME1, COX14, RPS28A and spike (0.4 µM) using the CFX384 Real-Time System (Biorad).

## Acknowledgements

The authors would like to acknowledge Tom Dever for his thoughtful and critical suggestions. This work was supported by the Intramural Research Program (JRL and AGH) of the National Institutes of Health (NIH), NIH grant GM62128 (previously to JRL), and by Wellcome Trust grant 090812/B/09/Z and the Centrum of Excellence of the Czech Science Foundation P305/12/G034 (both to LSV). CEA was further supported by an NIH minority supplement to NIH grant GM62128 and by a Leukemia and Lymphoma Society CDP Fellowship.

## Additional information

### Competing interests

AGH: Reviewing editor, *eLife*. The other authors declare that no competing interests exist.

### Funding

| Funder | Grant reference number | Author |
| --- | --- | --- |
| National Institutes of Health | Intramural Research Program | Colin Echeverría Aitken Wen-Ling Chiu Fujun Zhou Alan G Hinnebusch Jon R Lorsch |
| Leukemia and Lymphoma Society | 5199-12 | Colin Echeverría Aitken |
| Wellcome Trust | 090812/B/09/Z | Leoš Shivaya Valášek |
| Centrum of Excellence of the Czech Science Foundation | P305/12/G034 | Leoš Shivaya Valášek |
| National Institutes of Health | GM62128 | Jon R Lorsch |

The funders had no role in study design, data collection and interpretation, or the decision to submit the work for publication.

### Author contributions

CEA, Purified all the eIF3 variants with assistance from Petra and Wen-Ling. Together with Jon, Alan, and Leos, he conceived of all the in-vitro experiments, which he performed himself. Wrote the first draft of the manuscript, which he further revised and edited with Jon, Alan, and Leos. Conception and design, Acquisition of data, Analysis and interpretation of data, Drafting or revising the article; PB, Created the strains for expressing and purifying all eIF3 variants, with the exception of those created by Wen-Ling. Petra additionally helped with the purification of several eIF3 variants. Drafting or revising the article, Contributed unpublished essential data or reagents; VV, Performed the in vivo mRNA crosslinking experiments. Acquisition of data; W-LC, Created the strains for expressing and purifying H725P, R731I, and box6 eIF3, and helped with these purifications. Contributed unpublished essential data or reagents; FZ, Created the constructs for in-vitro transcription of the unstructured model mRNAs. Contributed unpublished essential data or reagents; LSV, AGH, JRL, Conception and design, Analysis and interpretation of data, Drafting or revising the article

Author ORCIDs

Alan G Hinnebusch, http://orcid.org/0000-0002-1627-8395

Jon R Lorsch, http://orcid.org/0000-0002-4521-4999

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

## Appendix 1

# Supplementary materials and methods

## Yeast strain constructions

A list of all strains used throughout this study can be found in *Appendix 1—table 1*.

To create PBH57 (*prb1Δ pep4Δ tif34Δ* YEp-TIF34) and PBH42 (*prb1Δ pep4Δ tif35Δ* YEp-TIF35), *LEU2*–based covering plasmids YEp111-MET-TIF34-L and YEp111-MET-TIF35-L, expressing *TIF34* or *TIF35* under the control of the methionine promoter, respectively, were first introduced into strain BJ5465 (*prb1Δ pep4Δ*) strain using the lithium acetate yeast transformation protocol (*Gietz and Woods, 2002*). The resulting transformants were first selected on SC media lacking leucine and methionine. Positively scoring clones were subsequently transformed with the *tif34Δ::hisG::URA3::hisG* (pTZ-tif34Δ) and *tif35Δ::hisG:: URA3::hisG* (B444) disruption cassettes, respectively, to delete a gene encoding a given eIF3 subunit. The Ura$^+$ colonies were selected on SC-URA plates and the uracil auxotrophy was subsequently regained by growing the transformants on medium containing 5-fluoroorotic acid (5-FOA). Successful completion of these genetic manipulations was verified by growing the resulting strains, auxotrophic for uracil, on plates supplemented with or lacking methionine. Only those cells where a chromosomal allele encoding a given eIF3 subunit was deleted, failed to grow on media containing methionine. Thus generated and verified strains were finally transformed with YEp-TIF34 and YEp-TIF35 and the leucine auxotrophy (a loss of the original covering plasmids YEp111-MET-TIF34-L and YEp111-MET-TIF35-L, respectively) was regained by growing the cells in liquid media containing leucine for several rounds of exponential growth and selecting for clones that grew in the absence of uracil but not in the absence of leucine in the media. The resulting strains were named PBH57 (*prb1Δ pep4Δ tif34Δ* YEp-TIF34) and PBH42 (*prb1Δ pep4Δ tif35Δ* YEp-TIF35).

To create PBH44 (*prb1Δ pep4Δ prt1Δ* YEpPRT1-U) and PBH56 (*prb1Δ pep4Δ nip1Δ* YEpNIP1-His-U), *TRP1*–based covering plasmids YCpAH-MET-PRT1-W and YCpMJ-MET-NIP1-W, expressing *PRT1* and *NIP1* under the control of the methionine promoter, respectively, were first introduced into the BJ5465 (*prb1Δ pep4Δ*) strain using the LiAc yeast transformation protocol. The resulting transformants were first selected on SC media lacking tryptophan and methionine. Positively scoring clones were subsequently transformed with the *prt1Δ::hisG::URA3::hisG* (B3289) and *nip1Δ::hisG::URA3::hisG* (pLV10) disruption cassettes, respectively, to delete a gene of a given eIF3 subunit. The URA+ colonies were selected on SC-URA plates and the uracil auxotrophy was subsequently regained by growing the transformants on 5-FOA containing plates. Successful completion of these genetic manipulations was verified by growing the resulting strains, auxotrophic for uracil, on plates supplemented with or lacking methionine. Only those cells where a chromosomal allele of a given eIF3 subunit was completely deleted failed to grow on media containing methionine. Thus generated and verified strains were finally transformed with YEpPRT1-U and YEpNIP1-His-U and the tryptophan auxotrophy (a loss of the original covering plasmids YCpAH-MET-PRT1-W and YCpMJ-MET-NIP1-W, respectively) was regained by growing the cells in liquid media containing tryptophan for several rounds of exponential growth and selecting for those clones that did grow in the absence of uracil but not in the absence of tryptophan in the media. The resulting strains were named PBH44 (*prb1Δ pep4Δ prt1Δ* YEpPRT1-U) and PBH56 (*prb1Δ pep4Δ nip1Δ* YEpNIP1-His-U).

To produce PBH65, PBH66 and PBH67 overexpressing all eIF3 subunits, PBH56 (*prb1Δ pep4Δ nip1Δ* YEpNIP1-His-U) was first subjected to double transformation with YEpPRT1-His-TIF34-TIF35-W/YEpTIF32-nip1-Δ60-L, YEpPRT1-His-TIF34-TIF35-W/Yep3TIF32-nip_GAP85-L or YEpPRT1-His-TIF34-TIF35-W/Yep3TIF32-nip_Box12-SPW-L, respectively. The resulting double transformants were selected on SC-LEU-TRP but +URA to minimize the number of revertants generated during this procedure for unknown reasons. The resident

*URA3*-based YEpNIP1-His-U plasmid carrying was contra-selected against on SC plates containing 5-FOA.

To produce PBH54 and PBH55 overexpressing all eIF3 subunits, PBH44 (*prb1Δ pep4Δ prt1Δ* YEpPRT1-U) was first subjected to double transformation with YEpprt1-rnp1-His-TIF34-TIF35-W/YEpTIF32-NIP1-L or with YEpprt1-1-His-TIF34-TIF35-W / YEpTIF32-NIP1-L, respectively. The resulting double transformants were selected on SC-LEU-TRP but +URA to minimize the number of revertants. The resident *URA3*-based YEpPRT1-U plasmid carrying was contra-selected against on SC plates containing 5-FOA.

To produce PBH51 and PBH143 overexpressing all eIF3 subunits, WLCY12 (*prb1Δ pep4Δ tif32Δ* YCp*TIF32*-His-U) was first subjected to double transformation with YEpPRT1-His-TIF34-TIF35-W/YEptif32-Δ8-NIP1-L or YEpPRT1-His-TIF34-TIF35-W/Yep3tif32_BOX37-NIP-L, respectively. The resulting double transformants were selected on SC-LEU-TRP but +URA to minimize the number of revertants. The resident *URA3*-based YCp*TIF32*-His-U plasmid carrying was contra-selected against on SC plates containing 5-FOA.

To produce PBH71 and PBH73 overexpressing all eIF3 subunits, PBH57 (*prb1Δ pep4Δ tif34Δ* YEp-TIF34) was first subjected to double transformation with YEpPRT1-His-tif34_DDKK-TIF35-W/YEpTIF32-NIP1-L or with YEpPRT1-His-tif34_Q258R-TIF35-W/YEpTIF32-NIP1-L, respectively. The resulting double transformants were selected on SC-LEU-TRP but +URA to minimize the number of revertants. The resident *URA3*-based YEp-TIF34 plasmid carrying was contra-selected against on SC plates containing 5-FOA.

To produce PBH76 overexpressing all eIF3 subunits, PBH42 (*prb1Δ pep4Δ tif35Δ* YEp-TIF35) was first subjected to double transformation with YEpPRT1-His-TIF34-tif35_KLF-W/YEpTIF32-NIP1-L. The resulting double transformants were selected on SC-LEU-TRP but +URA to minimize the number of revertants. The resident *URA3*-based YEp-TIF35 plasmid carrying was contra-selected against on SC plates containing 5-FOA.

To create strains WLCY11 and WLCY12, protease-deficient yeast strains BJ5464 (MATα *ura3-52 trp1 leu2-Δ1 his3-Δ200 pep4::HIS3 prb1-Δ1.6R can1 GAL*[+]) and BJ5465 (*MATa ura3-52 trp1 leu2- Δ1 his3-Δ200 pep4::HIS3 prb1-Δ1.6 can1 GAL*[+]) were each transformed with single-copy (sc) plasmid p3908/YCpTIF32-His-U (*TIF32 URA3*), respectively, and then with a PCR fragment containing the *tif32Δ::KanMX4* allele amplified from the appropriate deletion mutant from the *Saccharomyces* Genome Deletion Project (**Giaever et al., 2002**), purchased from Research Genetics, selecting for resistance to G418.

WLCY12 was transformed with high-copy (hc) plasmid p3127/yEp-PRT1His-TIF34HA-TIF35FL-Leu (*PRT1-His TIF34-HA TIF35-FLAG LEU2*) to create strain WLCY13.

To create strains WLCY14 to WLCY17, WLCY11 was transformed with pWCB23, pWCB24, pWCB25, and pWCB26, respectively, and the resident *TIF32/NIP1/URA3* plasmid was evicted on medium containing 5-fluoro-orotic acid (5-FOA).

To generate yeast strains WLCY18 to WLCY21 for over-expressing eIF3 subunits, haploid strains WLCY14 to WLCY17 were mated to WLCY13, selecting for diploids on SC-LWU plates, and the resident *TIF32/NIP1/URA3* plasmid was evicted using 5-FOA.

## Plasmid constructions

A list of all the plasmids and PCR primers used throughout this study can be found in *Appendix 1—tables 2* and *3*, respectively.

To insert the coding sequences for the 8xHis tag into the overexpressing vector YEpPRT1-TIF34-TIF35-U, the fusion PCR was performed using the following pairs of primers PRT1-SpeI – PRT1-8xHis-PstI-R and PRT1-8xHis-F – PRT1-SphI-R, with YEpPRT1-TIF34-TIF35-U as a template. The PCR products thus obtained were used in a 1:1 ratio as templates for a third PCR amplification with primers PRT1-SpeI and PRT1-SphI-R. The resulting PCR product was

digested with *Spe*I-*Sph*I and inserted into *Spe*I-*Sph*I-digested YEpPRT1-TIF34-TIF35-U producing YEpPRT1-His-TIF34-TIF35-U.

To produce YEpPRT1-His-TIF34-TIF35-W, *URA3* was replaced by *TRP1* by cutting YEpPRT1-His-TIF34-TIF35-U with *Ahd*I-*Sac*I and replacing the ~4 kb fragment with the ~3.7 kb long *Ahd*I-*Sac*I-digested fragment from YEp112.

YEpTIF32-NIP1-L was generated by PCR using primers MJRNIP1Sal-Xba and NIP1-noHis-BamHI-R, and YEpTIF32-NIP1-His-L as a template. The resulting PCR product was digested with *Xba*I-*Bam*HI and inserted into *Xba*I-*Bam*HI cleaved YEpTIF32-NIP1-His-L to produce YEpTIF32-NIP1-L.

YEpTIF32-nip1-Δ60-L was generated by PCR using primers MJRNIP1Sal-Xba and PBNIPD60-BamHI-R, and YEpTIF32-NIP1-L as a template. The resulting PCR product was digested with *Xba*I-*Bam*HI and inserted into *Xba*I-*Bam*HI cleaved YEpTIF32-NIP1-L.

YEpprt1-1-His-TIF34-TIF35-W was generated by PCR using primers PRT1-SpeI and PRT1-8xHis-PstI-R, and pLPY202 as a template. The resulting PCR product was digested with *Spe*I-*Pst*I and inserted into *Spe*I-*Pst*I cleaved YEpPRT1-His-TIF34-TIF35-W.

To produce YEpprt1-rnp1-His-TIF34-TIF35-W, ~ 2.3 kb fragment digested from p4473 with *Spe*I-*Ahd*I was inserted into *Spe*I-*Ahd*I cleaved YEpPRT1-His-TIF34-TIF35-W.

To produce YEptif32-Δ8-NIP1-L, ~ 3.2 kb fragment digested from YEptif32-Δ8-NIP1-His-L with *Msc*I-*Pst*I was inserted into *Msc*I-*Pst*I cleaved YEpTIF32-NIP1-L

To produce Yep3tif32_BOX37-NIP-L, fragment digested from YCp-a/tif32-Box37-H with *Msc*I-*Bsg*I was inserted into *Msc*I-*Bsg*I cleaved YEpTIF32-NIP1-L.

To produce Yep3TIF32-nip_GAP85-L, fragment digested from YCpNIP1-GAP85 with *Sca*I-*Xba*I was inserted into *Sca*I-*Xba*I cleaved YEpTIF32-NIP1-L.

To produce YEp3TIF32-nip_Box12-SPW-L, fragment digested from YCpNIP1-Box12-SPW with *Sca*I-*Xba*I was inserted into *Sca*I-*Xba*I cleaved YEpTIF32-NIP1-L.

YEpPRT1-His-tif34_Q258R-TIF35-W was generated by PCR using primers PBTIF34SacI and TIF34-SmaI-R, and YCpL-i/tif34-HA-3 as a template. The resulting PCR product was digested with *Sma*I-*Sac*I and inserted into *Sma*I-*Sac*I cleaved YEpPRT1-His-TIF34-TIF35-W.

YEpPRT1-His-tif34_DDKK-TIF35-W was generated by PCR using primers PBTIF34SacI and TIF34-SmaI-R, and YCp-i/TIF34-D207K-D224K-HA as a template. The resulting PCR product was digested with *Sma*I-*Sac*I and inserted into *Sma*I-*Sac*I cleaved YEpPRT1-His-TIF34-TIF35-W.

YEpPRT1-His-TIF34-tif35_KLF-W was generated by PCR using primers TIF35-SphI and TIF35-SmaI-R, and YCp22-g/TIF35-KLF as a template. The resulting PCR product was digested with *Sma*I-*Sph*I and inserted into *Sma*I-*Sph*I cleaved YEpPRT1-His-TIF34-TIF35-W.

YCpAH-MET-PRT1-W was generated by PCR using primers AH-PRT1-SalI and AH-PRT1-HindIII-R, and pGAD-PRT1 as a template. The resulting PCR product was digested with *Sal*I-*Bam*HI and inserted into *Sal*I-*Bam*HI cleaved YCplac22MET-W.

To produce YEp111-MET-TIF35-L, fragment containing *TIF35* digested from pGAD-TIF35 with *Bam*HI-*Pst*I and fragment containing MET3 promotor digested from YCpLV06 with *Bam*HI-*Eco*RI were inserted into *Pst*I-*Eco*RI cleaved YCplac111.

YEp111-MET-TIF34-L was generated by PCR using primers AH-PRT1-SalI and AH-PRT1-HindIII-R, and pGAD-TIF34 as a template. The resulting PCR product was digested with *Bam*HI-*Pst*I and inserted into *Bam*HI-*Pst*I cleaved YEp111-MET-TIF35-L.

YEpPRT1-U was created by inserting the *Hind*III-*Pst*I digested PCR product obtained with primers LVPRT1-5' and LVPRT1-3'R using genomic DNA obtained from yeast strain BY4741 as template into *Hind*III-*Pst*I digested YEplac195.

YEpPRT1-TIF35-U was created by inserting the *Sph*I-*Sal*I digested PCR product obtained with primers LVTIF35-5' and LVTIF35-3'R using genomic DNA obtained from yeast strain BY4741 as template into *Sph*I-*Sal*I digested YEpPRT1-U.

YEpPRT1-TIF34-TIF35-U was created by inserting the *Sma*I-*Sac*I digested PCR product obtained with primers LVTIF34-5' and LVTIF34-3'R using genomic DNA obtained from yeast strain BY4741 as template into *Sma*I-*Sac*I digested YEpPRT1-TIF35-U.

To produce YEpTIF32-NIP1-His-L, 4.7kbp-fragment containing *TIF32* digested from p3131 with *Sac*I-*Pst*I was inserted into *Sac*I-*Pst*I cleaved YEpNIP1-His-L.

To produce YEptif32-Δ8-NIP1-His-L, fragment digested from pRSTIF32-Δ8-His with *Msc*I-*Pst*I was inserted into *Msc*I-*Pst*I cleaved YEpTIF32-NIP1-His-L.

pWCB27, pWCB28, pWCB29 were made by inserting the *Pst*I-*Msc*I fragments from pWLCB01, p4577, and pRS-a/tif32-box6-His-L, respectively, into p3131 digested with *Pst*I and *Msc*I. Because *Pst*I cuts in the middle of *tif32-box6*, intact ORF DNA was obtained by partial digestion of pRS-a/tif32-box6-His-L.

pWCB23, pWCB24, pWCB25, and pWCB26, containing *TRP1*, were generated from p3131, pWCB27, pWCB28, pWCB29 by using the marker swap plasmid pUT11.

**Appendix 1—table 1.** Yeast strains used in this study.

| Strain | Genotype | Source or reference |
|---|---|---|
| BY4741 | *MATa his3Δ0 leu2Δ0 met15Δ0 ura3Δ0* | (**Brachmann et al., 1998**) |
| BJ5465 | *MATa ura3-52 trp1 leu2- Δ1 his3-Δ200 pep4::HIS3 prb1-Δ1.6 can1 GAL*+ | Elizabeth Jones |
| WLCY12 | *MATa ura3-52 trp1 leu2-Δ1 his3-Δ200 pep4::HIS3 prb1-Δ1.6 can1 GAL*+ *tif32Δ::kanMX4* p3908 sc[*TIF32, NIP1, URA3*] | This study |
| PBH42 | *MATa ura3-52 trp1 leu2-Δ1 his3-Δ200 pep4::HIS3 prb1-Δ1.6R can1 TIF35-del GAL+* (YEp-TIF35) | This study |
| PBH44 | *MATa ura3-52 trp1 leu2-Δ1 his3-Δ200 pep4::HIS3 prb1-Δ1.6R can1 prt1-Δ GAL+* (YEpPRT1-U) | This study |
| PBH51 | *MATa ura3-52 trp1 leu2-Δ1 his3-Δ200 pep4::HIS3 prb1-Δ1.6R can1 GAL+ tif32Δ::KanMX4* (YEpPRT1-His-TIF34-TIF35-W, YEptif32-Δ8-NIP1-L) | This study |
| PBH143 | *MATa ura3-52 trp1 leu2-Δ1 his3-Δ200 pep4::HIS3 prb1-Δ1.6R can1 GAL+ tif32Δ::KanMX4* (YEpPRT1-His-TIF34-TIF35-W, Yep3tif32_BOX37-NIP-L) | This study |
| PBH54 | *MATa ura3-52 trp1 leu2-Δ1 his3-Δ200 pep4::HIS3 prb1-Δ1.6R can1 prt1-Δ GAL+* (YEpprt1-rnp1-His-TIF34-TIF35-W, YEpTIF32-NIP1-L) | This study |
| PBH55 | *MATa ura3-52 trp1 leu2-Δ1 his3-Δ200 pep4::HIS3 prb1-Δ1.6R can1 prt1-Δ GAL+* (YEpprt1-1-His-TIF34-TIF35-W, YEpTIF32-NIP1-L) | This study |
| PBH56 | *MATa ura3-52 trp1 leu2-Δ1 his3-Δ200 pep4::HIS3 prb1-Δ1.6R can1 nip1-Δ GAL+* (YEpNIP1-His-U) | This study |
| PBH57 | *MATa ura3-52 trp1 leu2-Δ1 his3-Δ200 pep4::HIS3 prb1-Δ1.6R can1 tif34-Δ GAL+* (YEp-TIF34) | This study |
| PBH65 | *MATa ura3-52 trp1 leu2-Δ1 his3-Δ200 pep4::HIS3 prb1-Δ1.6R can1 nip1-Δ GAL+* (YEpPRT1-His-TIF34-TIF35-W, YEpTIF32-nip1-Δ60-L) | This study |
| PBH66 | *MATa ura3-52 trp1 leu2-Δ1 his3-Δ200 pep4::HIS3 prb1-Δ1.6R can1 nip1-Δ GAL+* (YEpPRT1-His-TIF34-TIF35-W, Yep3TIF32-nip_GAP85-L) | This study |
| PBH67 | *MATa ura3-52 trp1 leu2-Δ1 his3-Δ200 pep4::HIS3 prb1-Δ1.6R can1 nip1-Δ GAL+* (YEpPRT1-His-TIF34-TIF35-W, Yep3TIF32-nip_Box12-SPW-L) | This study |
| PBH71 | *MATa ura3-52 trp1 leu2-Δ1 his3-Δ200 pep4::HIS3 prb1-Δ1.6R can1 tif34-Δ GAL+* (YEpPRT1-His-tif34_DDKK-TIF35-W, YEpTIF32-NIP1-L) | This study |
| PBH73 | *MATa ura3-52 trp1 leu2-Δ1 his3-Δ200 pep4::HIS3 prb1-Δ1.6R can1 tif34-Δ GAL+* (YEpPRT1-His-tif34_Q258R-TIF35-W, YEpTIF32-NIP1-L) | This study |

*Appendix 1—table 1 continued*

| Strain | Genotype | Source or reference |
|---|---|---|
| PBH76 | MATa ura3-52 trp1 leu2-Δ1 his3-Δ200 pep4::HIS3 prb1-Δ1.6R can1 TIF35-Δ GAL+ (YEpPRT1-His-TIF34-tif35_KLF-W, YEpTIF32-NIP1-L) | This study |
| WLCY11 | *MATα ura3-52 trp1 leu2-Δ1 his3-Δ200 pep4::HIS3 prb1-Δ1.6R can1 GAL+ tif32Δ::kanMX4 p3908 sc[TIF32, NIP1, URA3]* | This study |
| WLCY13 | *MATa ura3-52 trp1 leu2-Δ1 his3-Δ200 pep4::HIS3 prb1-Δ1.6 can1 GAL+ tif32Δ::kanMX4 p3908 sc[TIF32, NIP1, URA3] p3127 hc[PRT1-His, TIF34-HA, TIF35-FLAG, LEU2]* | This study |
| WLCY14 | *MATα ura3-52 trp1 leu2-Δ1 his3-Δ200 pep4::HIS3 prb1-Δ1.6R can1 GAL+ tif32Δ::kanMX4 p5237 hc[TIF32, NIP1, TRP1]* | This study |
| WLCY15 | *MATα ura3-52 trp1 leu2-Δ1 his3-Δ200 pep4::HIS3 prb1-Δ1.6R can1 GAL+ tif32Δ::kanMX4 p5238 hc[tif32-H725P, NIP1, TRP1]* | This study |
| WLCY16 | *MATα ura3-52 trp1 leu2-Δ1 his3-Δ200 pep4::HIS3 prb1-Δ1.6R can1 GAL+ tif32Δ::kanMX4 p5239 hc[tif32-R731I, NIP1, TRP1]* | This study |
| WLCY17 | *MATα ura3-52 trp1 leu2-Δ1 his3-Δ200 pep4::HIS3 prb1-Δ1.6R can1 GAL+ tif32Δ::kanMX4 p5240 hc[tif32-box6, NIP1, TRP1]* | This study |
| WLCY18 | *MATa/MATα ura3-52/ura3-52 trp1/trp1 leu2-Δ1/leu2-Δ1 his3-Δ200/his3-Δ200 pep4::HIS3/pep4::HIS3 prb1-Δ1.6R/prb1-Δ1.6R can1/can1 GAL+/GAL+ tif32Δ::kanMX4/tif32Δ::kanMX p5237 hc[TIF32, NIP1, TRP1] p3127 hc[PRT1-His, TIF34-HA, TIF35-FLAG, LEU2]* | This study |
| WLCY19 | *MATa/MATα ura3-52/ura3-52 trp1/trp1 leu2-Δ1/leu2-Δ1 his3-Δ200/his3-Δ200 pep4::HIS3/pep4::HIS3 prb1-Δ1.6R/prb1-Δ1.6R can1/can1 GAL+/GAL+ tif32Δ::kanMX4/tif32Δ::kanMX p5238 hc[tif32-H725P, NIP1, TRP1] p3127 hc[PRT1-His, TIF34-HA, TIF35-FLAG, LEU2]* | This study |
| WLCY20 | *MATa/MATα ura3-52/ura3-52 trp1/trp1 leu2-Δ1/leu2-Δ1 his3-Δ200/his3-Δ200 pep4::HIS3/pep4::HIS3 prb1-Δ1.6R/prb1-Δ1.6R can1/can1 GAL+/GAL+ tif32Δ::kanMX4/tif32Δ::kanMX p5239 hc[tif32-R731I, NIP1, TRP1] p3127 hc[PRT1-His, TIF34-HA, TIF35-FLAG, LEU2]* | This study |
| WLCY21 | *MATa/MATα ura3-52/ura3-52 trp1/trp1 leu2-Δ1/leu2-Δ1 his3-Δ200/his3-Δ200 pep4::HIS3/pep4::HIS3 prb1-Δ1.6R/prb1-Δ1.6R can1/can1 GAL+/GAL+ tif32Δ::kanMX4/tif32Δ::kanMX p5240 hc[tif32-box6, NIP1, TRP1] p3127 hc[PRT1-His, TIF34-HA, TIF35-FLAG, LEU2]* | This study |

**Appendix 1—table 2.** Plasmids used in this study. High-copy, low-copy, and single-copy plasmids are designated hc, lc. and sc, respectively.

| Plasmid | Description | Source of reference |
|---|---|---|
| pLV10 | hc vector containing *nip1Δ::hisG::URA3::hisG* disruption cassette | (*Valasek et al., 2004*) |
| p4473 (pRS315-prt1-rnp1-His) | lc *LEU2* vector containing *prt1-rnp1-His* | (*Nielsen et al., 2006*) |
| pTZ-tif34Δ | hc vector containing *tif34Δ::hisG::URA3::hisG* disruption cassette | (*Asano et al., 1998*) |
| B3289 | hc vector containing *prt1Δ::hisG::URA3::hisG* disruption cassette | (*Nielsen et al., 2004*) |
| B444 | hc vector containing *tif35Δ::hisG::URA3::hisG* disruption cassette | (*Cuchalova et al., 2010*) |
| YCpMJ-MET-NIP1-W | sc *TRP1* vector containing *NIP1* under *MET3* promotor | (*Karásková et al., 2012*) |
| YCpAH-MET-PRT1-W | sc *TRP1* vector containing *PRT1* under *MET3* promotor | This study |
| pGAD-PRT1 | *PRT1* ORF cloned into pGAD424 | (*Asano et al., 1998*) |
| YCplac22MET-W | sc cloning vector with conditional *MET3* promoter, *TRP1* plasmid from YCplac22 | K. Nasmyth |

*Appendix 1—table 2 continued on next page*

**Appendix 1—table 2 continued**

| Plasmid | Description | Source of reference |
|---|---|---|
| pLPY202 | lc *URA3* vector containing *prt1-1*-His | (*Phan et al., 2001*) |
| YEp112 | hc *TRP1* vector | (*Gietz and Sugino, 1988*) |
| YEp111-MET-TIF34-L | hc *LEU2* vector containing *TIF34* under *MET3* promotor | This study |
| pGAD-TIF34 | *TIF34* ORF cloned into pGAD424 | (*Asano et al., 1998*) |
| YCpLV06 | sc *LEU2* vector containing *TIF32* under *MET3* promotor | (*Kovarik et al., 1998*) |
| YCplac111 | sc *LEU2* vector | (*Gietz and Sugino, 1988*) |
| pGAD-TIF35 | *TIF35* ORF cloned into pGAD424 | (*Asano et al., 1998*) |
| YEp111-MET-TIF35-L | hc *LEU2* vector containing *TIF35* under *MET3* promotor | This study |
| YEpNIP1-His-U | hc *URA3* vector containing *NIP1-His* | (*Valášek et al., 2002*) |
| YEpprt1-1-His-TIF34-TIF35-W | hc *TRP1* vector containing *prt1-1-His*, *TIF34* and *TIF35* | This study |
| YEpPRT1-His-TIF34-TIF35-W | hc *TRP1* vector containing *PRT1-His*, *TIF34* and *TIF35* | This study |
| YEpprt1-rnp1-His-TIF34-TIF35-W | hc TRP1 vector containing *prt1-rnp1-His*, *TIF34* and *TIF35* | This study |
| YEpPRT1-His-TIF34-TIF35-U | hc *URA3* vector containing *PRT1-His*, *TIF34* and *TIF35* | This study |
| YEpPRT1-U | hc *URA3* vector containing *PRT1* | This study |
| YEplac195 | hc *URA3* vector | (*Gietz and Sugino, 1988*) |
| YEpPRT1-TIF35-U | hc *URA3* vector containing *PRT1* and *TIF35* | This study |
| YEpPRT1-TIF34-TIF35-U | hc *URA3* vector containing *PRT1*, *TIF34* and *TIF35* | This study |
| YEpTIF32-NIP1-L | hc *LEU2* vector containing *TIF32* and *NIP1* | This study |
| YEpTIF32-NIP1-His-L | hc *LEU2* vector containing *TIF32* and *NIP1-His* | This study |
| YEpNIP1-His-L | hc *LEU2* vector containing *NIP1-His* | (*Valášek et al., 2002*) |
| YEpTIF32-nip1-Δ60-L | hc *LEU2* vector containing *TIF32* and *NIP1* truncated by 60 amino acid residues in its CTD | This study |
| YEptif32-Δ8-NIP1-His-L | hc *LEU2* vector containing *TIF32* with the deletion of first 200aa and *NIP1-His* | This study |
| pRSTIF32-Δ8-His | lc *TIF32-Δ8-His*[200–964], *LEU2* plasmid from pRS315 | (*Valášek et al., 2002*) |
| YEptif32-Δ8-NIP1-L | hc *LEU2* vector containing *TIF32* truncated by 200 amino acid residues in its NTD and *NIP1* | This study |
| YEp-TIF34 | hc *URA3* vector containing *TIF34* | (*Asano et al., 1998*) |
| YEp-TIF35 | hc *URA3* vector containing *TIF35* | (*Asano et al., 1998*) |
| Yep3tif32_BOX37-NIP-L | hc *LEU2* vector containing *TIF32* with ten alanine substitution in its NTD and *NIP1* | This study |
| YCp-a/tif32-Box37-H | sc *tif32-Box37-His* in *LEU2* plasmid, from YCplac111 | (*Khoshnevis et al., 2014*) |
| YEp3TIF32-nip_Box12-SPW-L | hc *LEU2* vector containing *TIF32* and *NIP1* containing K113S/K116P/K118W substitutions in Box12 | This study |
| YCpNIP1-Box12-SPW | sc *NIP1-His* containing K113S/K116P/K118W substitutions in Box12; *LEU2* plasmid from YCplac111 | (*Karásková et al., 2012*) |
| Yep3TIF32-nip_GAP85-L | hc *LEU2* vector containing *TIF32* and *NIP1* containing deletion of 85 residues (Val$^{60}$- Asn144) | This study |

*Appendix 1—table 2 continued on next page*

*Appendix 1—table 2 continued*

| Plasmid | Description | Source of reference |
|---|---|---|
| YCpNIP1-GAP85 | sc *NIP1-His* containing deletion of 85 residues (Val$^{60}$-Asn$^{144}$); *LEU2* plasmid from YCplac111 | {Karaskova:2012ce} |
| YEpPRT1-His-tif34_Q258R-TIF35-W | hc *TRP1* vector containing *PRT1-His*, *tif34-Q258R* and *TIF35* | This study |
| YCpL-i/tif34-HA-3 | sc *tif34-HA-Q258R* in *LEU2* plasmid from YCplac111 | (*Cuchalova et al., 2010*) |
| YEpPRT1-His-tif34_DDKK-TIF35-W | hc *TRP1* vector containing *PRT1-His*, *TIF34* containing D207K and D224K mutations and *TIF35* | This study |
| YCp-i/TIF34-D207K-D224K-HA | sc *TIF34-HA* containing D207K and D224K mutations in *LEU2* plasmid from YCplac111 | (*Herrmannová et al., 2012*) |
| YEpPRT1-His-TIF34-tif35_KLF-W | hc TRP1 vector containing *PRT1-His*, *TIF34* and *tif35-KLF* | This study |
| YCp22-g/TIF35-KLF | sc *TIF35-KLF-His* in*TRP1* plasmid from YCplac22 | (*Cuchalova et al., 2010*) |
| pWCB27 | hc *NIP1, tif32-H725P, URA3* plasmid, from YEplac195 | This study |
| pWCB28 | hc *NIP1, tif32-R731I, URA3* plasmid, from YEplac195 | This study |
| pWCB29 | hc *NIP1, tif32-box6, URA3* plasmid, from YEplac195 | This study |
| pWLCB01 | *tif32-H725P-His* in lc *LEU2* plasmid, from pRS315 | (*Chiu et al., 2010*) |
| p4577 (pRS-a/tif32-R731I-His-L) | *tif32-R731I-His* in lc *LEU2* plasmid, from pRS315 | (*Chiu et al., 2010*) |
| pRS-a/tif32-box6-His-L | *tif32-box6-His* in lc *LEU2* plasmid, from pRS315 | (*Chiu et al., 2010*) |
| p3131 (pLPY-NIP1-TIF32) | hc *NIP1, TIF32, URA3* plasmid, from YEplac195 | (*Phan et al., 2001*) |
| p3908 (YCp-a/TIF32-His-U) | sc *TIF32-His URA3* plasmid | (*Valášek et al., 2003*) |
| pWCB23 | hc *NIP1, TIF32, TRP1* plasmid, from YEplac195 | This study |
| pWCB24 | hc *NIP1, tif32-H725P, TRP1* plasmid, from YEplac195 | This study |
| pWCB25 | hc *NIP1, tif32-R731I, TRP1* plasmid, from YEplac195 | This study |
| pWCB26 | hc *NIP1, tif32-box6, TRP1* plasmid, from YEplac195 | This study |
| pUT11 | *URA3* to *TRP1* marker swap plasmid | (*Cross, 1997*) |
| p3127 (pLPY-PRT1His-TIF34HA-TIF35FL-Leu) | hc *PRT1-His, TIF34-HA, TIF35-FLAG, LEU2* plasmid from YEplac 181 | (*Phan et al., 2001*) |

**Appendix 1—table 3.** Primers used in this study.

| Primer name | Primer sequence (5´to 3´) |
|---|---|
| MJNIP1Sal-Xba | AGCAAAGAGTCAAGAAAGTTTCTA |
| NIP1-noHis-BamHI-R | TTATTGGATCCTCAACGACGATTTGATGGTGGGTTAAGACG |
| PBNIPD60-BamHI-R | CCCCGGATCCTCACACCTTATTTTCTGGAAGATC |
| PRT1-8xHis-F | ATTTCATGAACTTACGGGCTTGTATGTAAA |
| PRT1-8xHis-PstI-R | TTTACATACAAGCCCGTAAGTTCATGAAATCTGCAGTTAGTGGTGGTGGTGGTGGTGGTGGTGTTCGACCTTTTCCTTTGTTTCTTCCAAAAC |
| PRT1-SpeI | GACGTGAAGACTAGTGTGTTC |
| PRT1-SphI-R | TGCGTCTACTTGTGCATGCAT |
| PBTIF34SacI | AGTGAATTCGAGCTCCTTATTCAGCGG |
| TIF34-SmaI-R | TTATTCCCGGGGTTAATTAGCTTCTTGCATGTGCTC |
| TIF35-SphI | AATAAGCATGCACAAGTAGACGCACCTAAAAG |

*Appendix 1—table 3 continued on next page*

*Appendix 1—table 3 continued*

| Primer name | Primer sequence (5′to 3′) |
|---|---|
| TIF35-SmaI-R | TTATTCCCGGGCTATTCCTTAACCTTAGGTTTGGA |
| AH-PRT1-SalI | ATATATGTCGACATGACTACCGAGACTTTCGAA |
| AH-PRT1-HindIII-R | GTCCAAAGCTTCTGAATAAGCCCT |
| OJ-TIF34 | attcggatccatatgaaggctatcaaattaacaggt |
| OJ-TIF34-R | atctctgcagttaattagcttcttgcatgtgctcttt |
| LVPRT1-5′ | aataaAAGCTTAGGGCGATCTGCTACAGGAAGCTA |
| LVPRT1-3′R | aataaCTGCAGGCATGCATACAAAGATAATAGAGCCTATT |
| LVTIF35-5′ | aataaGCATGCACAAGTAGACGCACCTAAAAGTCC |
| LVTIF35-3′R | aataaGTCGACACATATTCACGACAGCCTCTGAGC |
| LVTIF34-5′ | aataaGAGCTCAAActcgagGAGCTGATAAAACCCTACACTACGGTGTAA |
| LVTIF34-3′R | CGCCTATGCCCGGGAACTGCATAC |

