## [Decision Letter]

Thank you for submitting your article "eIF3 plays distinct roles at the mRNA entry and exit channels of the ribosomal preinitiation complex" for consideration by *eLife*. Your article has been reviewed by two peer reviewers, and the evaluation has been overseen by a Reviewing Editor and Kevin Struhl as the Senior Editor. The reviewers have opted to remain anonymous.

The reviewers have discussed the reviews with one another and the Reviewing Editor has drafted this decision to help you prepare a revised submission.

Summary:

The authors use a reconstituted translation initiation system from yeast to test the functional contributions of eIF3 to initiation. In yeast, eIF3 is composed of 5 core subunits that have been studied genetically and biochemically, primarily by the authors and their collaborators, yielding a number of important insights into translation initiation. The reconstituted system used here allows the authors to more rigorously explore eIF3 function as compared to earlier efforts. The authors purify a series of mutant forms of eIF3 that had been studied before (Figure 1—figure supplement 2) to identify biochemically the specific defects in 43S preinitiation complex (43S PIC) and 48S PIC assembly. They uncover a requirement for eIF3a NTD in stabilizing mRNA interactions at the exit channel. Their results also indicate that the eIF3 entry channel arm and eIF3b stabilize TC binding to the pre-initiation complex and is important for mRNA recruitment.

The authors show that the various mutant forms of eIF3 have little impact on binding events, i.e. eIF3 binding to the 43S and 48S PIC, or eIF2/GTP/Met-tRNAi ternary complex (TC) binding to the 43S PIC. The authors do see evidence for some cooperativity in eIF3 binding (Figure 2—figure supplement 2). From their results, the authors also suggest that some of the modulation could be due to shifts in the equilibrium between an "open" and "closed" form of the PIC seen in prior biochemical and structural work.

Essential revisions:

The reviewers thought that overall this is a solid, carefully conducted biochemical study. The work is thorough and comprehensive and the experiments are well performed. The authors are also cautious in their interpretations. However, they raised a few concerns about the novelty and interpretation of the results that need to be addressed as follows:

With respect to the KLF and Q258R mutations, there is nothing new here, and actually less than in the original paper describing those mutants (Cuchalova, 2010). It is important to clarify what they have actually studied here. The KLF and Q258R mutations are unremarkable in all of the assays here, yet are fascinating in the in vivo context. Their effect arises during scanning, i.e. after assembly of the initial 48S PIC. The delta-8 mutation also affects scanning, but as indicated in the earlier work is distinct from the effects of the KLF and Q258R mutations. Here, delta-8 has a dramatic effect on mRNA recruitment, revealed only with mRNAs truncated at entry site (Figure 5), whereas the KLF and Q258R mutations don't really have any effect in this paper.

This raises an important question as to whether the assay used here reports on scanning (authors' statement in the fifth paragraph of the Introduction). It probably does not, otherwise they would see defects with the KLF and Q258R mutations. Thus, the authors need to reword the Abstract and Discussion to indicate that their present reconstituted system only reports on steps not dependent on scanning.

Based on these comments we request you to rewrite of the Abstract and Discussion. This would help the authors bring out what's new, which now would be lost on most readers.

---

## [Author Response]

*[…] Essential revisions:*

*The reviewers thought that overall this is a solid, carefully conducted biochemical study. The work is thorough and comprehensive and the experiments are well performed. The authors are also cautious in their interpretations. However, they raised a few concerns about the novelty and interpretation of the results that need to be addressed as follows:*

*With respect to the KLF and Q258R mutations, there is nothing new here, and actually less than in the original paper describing those mutants (Cuchalova, 2010). It is important to clarify what they have actually studied here. The KLF and Q258R mutations are unremarkable in all of the assays here, yet are fascinating in the in vivo context. Their effect arises during scanning, i.e. after assembly of the initial 48S PIC. The delta-8 mutation also affects scanning, but as indicated in the earlier work is distinct from the effects of the KLF and Q258R mutations. Here, delta-8 has a dramatic effect on mRNA recruitment, revealed only with mRNAs truncated at entry site (Figure 5), whereas the KLF and Q258R mutations don't really have any effect in this paper.*

*This raises an important question as to whether the assay used here reports on scanning (authors' statement in the fifth paragraph of the Introduction). It probably does not, otherwise they would see defects with the KLF and Q258R mutations. Thus, the authors need to reword the Abstract and Discussion to indicate that their present reconstituted system only reports on steps not dependent on scanning.*

*Based on these comments we request you to rewrite of the Abstract and Discussion. This would help the authors bring out what's new, which now would be lost on most readers.*

We agree with the reviewers that we do not yet have definitive evidence that our mRNA recruitment assay reports on scanning with the mRNAs used in these studies and we have altered the text in the Introduction, Results and Discussion to reflect this. It should be noted, however, that recruitment of the capped, natural mRNA RPL41A that we used in this work is highly dependent on eIF4A and ATP, which is consistent with an active, energy-dependent process that could include scanning. We have now emphasized this point as well (Results: first paragraph).

Although we do not observe significant effects on the kinetics of *RPL41A* recruitment in the presence of the *Q258R* eIF3i and *KLF* eIF3g mutations, we nonetheless observe severe defects in both the rate and extent of mRNA recruitment in the simultaneous absence of both subunits (a/b/c subcomplex). Together with the previous observation that the *DDKK* eIF3i mutation (which evicts eIF3i and eIF3g from the eIF3 complex) confers a phenotype consistent with a severe scanning arrest defect in vivo, we believe this is consistent with the interpretation that our assay (which depends on the presence of an AUG codon in the mRNA) reports on events up to and including start codon recognition, and is thus potentially sensitive to scanning defects as well. It is possible that the recruitment of *RPL41A* mRNA (whose 5’ UTR is relatively short, at 22 nt, and lacks highly-structured elements) is not rate-limited by the scanning defects of the *Q258R* eIF3i and *KLF* eIF3g mutations, whose in vivo phenotypes suggest defects in rate or processivity of scanning through secondary structures, respectively. These defects might become rate-limiting for the recruitment of natural mRNAs whose 5’-UTRs are longer or more highly structured.

We thank the reviewers for pointing out the apparent opposition of these observations, and we have added text to both the Results and Discussion sections, shown below to clarify them. Because we did not make any claims in the Abstract regarding the effects of eIF3 mutations on scanning, and because it remains entirely possible that our in vitro assay does in fact report on scanning defects (for the reasons given above), we chose not to exclude this possibility in the Abstract. Even if it were true, it would not change any of the essential conclusions described in the Abstract.

Additions to Results:

“The observed 48S complex band is dependent on the presence of an AUG codon in themRNA (Mitchell et al., 2010)(Figure 4—figure supplement 1), and recruitment of capped *RPL41A* mRNA is highly dependent on the presence of both eIF4A and ATP in this in vitro system […] Nonetheless, the fact that both the extent and rate of recruitment of *RPL41A* mRNA are severely compromised when eIF3 is replaced with the a/b/c sub-complex, together with the previous observation that the loss of eIF3i and eIF3g from the eIF3 complex in vivo critically compromises scanning arrest (Herrmannová et al., 2012), is consistent with our interpretation that this assay is sensitive to scanning defects.”

Addition to Discussion:

“In light of the severe mRNA recruitment defects we observe in the simultaneous absence of eIF3i and eIF3g, it is notable that neither the *Q258R* eIF3i mutation nor the *KLF* eIF3g mutation confer significant defects in mRNA recruitment. […] It will be interesting to see the effects of these and other mutations that affect the efficiency of scanning by the PIC on the in vitro recruitment of natural mRNAs with longer, more structured 5’ UTRs.”